# Structural insights into Cullin4-RING ubiquitin ligase remodelling by Vpr from simian immunodeficiency viruses

Sofia Banchenko[1☉], Ferdinand Krupp[1☉], Christine Gotthold[1], Jörg Bürger[1,2], Andrea Graziadei[3], Francis J. O'Reilly[3], Ludwig Sinn[3], Olga Ruda[1], Juri Rappsilber[3,4], Christian M. T. Spahn[1], Thorsten Mielke[2], Ian A. Taylor[5], David Schwefel[1]*

1 Charité–Universitätsmedizin Berlin, corporate member of Freie Universität Berlin and Humboldt-Universität zu Berlin, Institute of Medical Physics and Biophysics, Berlin, Germany, 2 Microscopy and Cryo-Electron Microscopy Service Group, Max-Planck-Institute for Molecular Genetics, Berlin, Germany, 3 Bioanalytics Unit, Institute of Biotechnology, Technische Universität Berlin, Berlin, Germany, 4 Wellcome Centre for Cell Biology, University of Edinburgh, Edinburgh, United Kingdom, 5 Macromolecular Structure Laboratory, The Francis Crick Institute, London, United Kingdom

☉ These authors contributed equally to this work.
* david.schwefel@charite.de

**Data Availability Statement:** The coordinates and structure factors for the crystal structures have been deposited at the Protein Data Bank (PDB, https://www.rcsb.org/) with the accession codes

## Abstract

Viruses have evolved means to manipulate the host's ubiquitin-proteasome system, in order to down-regulate antiviral host factors. The Vpx/Vpr family of lentiviral accessory proteins usurp the substrate receptor DCAF1 of host Cullin4-RING ligases (CRL4), a family of modular ubiquitin ligases involved in DNA replication, DNA repair and cell cycle regulation. CRL4$^{DCAF1}$ specificity modulation by Vpx and Vpr from certain simian immunodeficiency viruses (SIV) leads to recruitment, poly-ubiquitylation and subsequent proteasomal degradation of the host restriction factor SAMHD1, resulting in enhanced virus replication in differentiated cells. To unravel the mechanism of SIV Vpr-induced SAMHD1 ubiquitylation, we conducted integrative biochemical and structural analyses of the Vpr protein from SIVs infecting *Cercopithecus cephus* (SIV$_{mus}$). X-ray crystallography reveals commonalities between SIV$_{mus}$ Vpr and other members of the Vpx/Vpr family with regard to DCAF1 interaction, while cryo-electron microscopy and cross-linking mass spectrometry highlight a divergent molecular mechanism of SAMHD1 recruitment. In addition, these studies demonstrate how SIV$_{mus}$ Vpr exploits the dynamic architecture of the multi-subunit CRL4$^{DCAF1}$ assembly to optimise SAMHD1 ubiquitylation. Together, the present work provides detailed molecular insight into variability and species-specificity of the evolutionary arms race between host SAMHD1 restriction and lentiviral counteraction through Vpx/Vpr proteins.

## Author summary

Due to the limited size of virus genomes, virus replication critically relies on host cell components. In addition to the host cell's energy metabolism and its DNA replication and

6ZUE (DDB1/DCAF1-CtD) and 6ZX9 (DDB1/DCAF1-CtD/T4L-Vprmus 1-92) (https://doi.org/10.2210/pdb6ZUE/pdb and https://doi.org/10.2210/pdb6ZX9/pdb). Cryo-EM reconstructions have been deposited at the Electron Microscopy Data Bank (EMDB,https://www.ebi.ac.uk/pdbe/emdb/) with the accession codes EMD-10611 (core), EMD-10612 (conformational state-1), EMD-10613 (state-2) and EMD-10614 (state-3) (https://www.ebi.ac.uk/pdbe/entry/emdb/EMD-10611 and https://www.ebi.ac.uk/pdbe/entry/emdb/EMD-10612 and https://www.ebi.ac.uk/pdbe/entry/emdb/EMD-10613 and https://www.ebi.ac.uk/pdbe/entry/emdb/EMD-10614). CLMS data have been deposited at the PRIDE database and can be accessed via the following link: https://www.ebi.ac.uk/pride/archive/projects/PXD020453.

**Funding:** This research was supported by the German Research Foundation (DFG) Emmy Noether Programme SCHW1851/1-1 (D.S.), the DFG project grant 329673113 (J.R.), the DFG cluster of excellence EXC 2008 - 390540038 - UniSysCat (L.S., J.R., C.M.T.S.), the DFG research training group GRK 2473 - 392923329 - Bioactive Peptides - Innovative Aspects of Synthesis and Biosynthesis (L.S.) https://www.dfg.de/, by an European Molecular Biology Organization (EMBO) Advanced laboratory start-up grant aALTF-1650 (D.S.) https://www.embo.org/, by Wellcome Trust grants 108014/Z/15/Z (I.A.T.) and 103139 (J.R.), by the Wellcome Centre for Cell Biology, which is supported by core funding from the Wellcome Trust 203149 (J.R.) https://wellcome.org/, and by an iNEXT instrumentation grant 3825 (D.S.) http://www.inext-eu.org/. The funders had no role in study design, data collection and analysis, decision to publish, or preparation of the manuscript.

**Competing interests:** The authors have declared that no competing interests exist.

protein synthesis apparatus, the protein degradation machinery is an attractive target for viral re-appropriation. Certain viral factors divert the specificity of host ubiquitin ligases to antiviral host factors, in order to mark them for destruction by the proteasome, to lift intracellular barriers to virus replication. Here, we present molecular details of how the simian immunodeficiency virus accessory protein Vpr interacts with a substrate receptor of host Cullin4-RING ubiquitin ligases, and how this interaction redirects the specificity of Cullin4-RING to the antiviral host factor SAMHD1. The studies uncover the mechanism of Vpr-induced SAMHD1 recruitment and subsequent ubiquitylation. Moreover, by comparison to related accessory proteins from other immunodeficiency virus species, we illustrate the surprising variability in the molecular strategies of SAMHD1 counteraction, which these viruses adopted during evolutionary adaptation to their hosts. Lastly, our work also provides deeper insight into the inner workings of the host's Cullin4-RING ubiquitylation machinery.

## Introduction

A large proportion of viruses have evolved means to co-opt their host's ubiquitylation machinery, in order to improve replication conditions, either by introducing viral ubiquitin ligases and deubiquitinases, or by modification of host proteins involved in ubiquitylation [1–3]. In particular, host ubiquitin ligases are a prominent target for viral usurpation, to redirect specificity towards antiviral host restriction factors. This results in recruitment of restriction factors as non-endogenous *neo*-substrates, inducing their poly-ubiquitylation and subsequent proteasomal degradation [4–8]. This counteraction of the host's antiviral repertoire is essential for virus infectivity and spread [9–12], and mechanistic insights into these specificity changes extend our understanding of viral pathogenesis and might pave the way for novel treatments.

Frequently, virally encoded modifying proteins associate with, and adapt the Cullin4-RING ubiquitin ligases (CRL4) [5]. CRL4 consists of a Cullin4 (CUL4) scaffold that bridges the catalytic RING-domain subunit ROC1 to the adaptor protein DDB1, which in turn binds to exchangeable substrate receptors (DCAFs, DDB1- and CUL4-associated factors) [13–17]. In some instances, the DDB1 adaptor serves as an anchor for virus proteins, which then act as "viral DCAFs" to recruit the antiviral substrate. Examples are the simian virus 5 V protein and mouse cytomegalovirus M27, which bind to DDB1 and recruit STAT1/2 proteins for ubiquitylation, in order to interfere with the host's interferon response [18–20]. Similarly, CUL4-dependent downregulation of STAT signalling is important for West Nile Virus replication [21]. In addition, the hepatitis B virus X protein hijacks DDB1 to induce proteasomal destruction of the structural maintenance of chromosome (SMC) complex to promote virus replication [22,23].

Viral factors also bind to and modify DCAF receptors in order to redirect them to antiviral substrates. Prime examples are the lentiviral accessory proteins Vpr and Vpx. All contemporary human and simian immunodeficiency viruses (HIV/SIV) encode Vpr, while only two lineages, represented by HIV-2 and SIV infecting mandrills, carry Vpx [24]. Vpr and Vpx proteins are packaged into progeny virions and released into the host cell upon infection, where they bind to DCAF1 [25]. In this work, corresponding simian immunodeficiency virus Vpx/Vpr proteins will be indicated with their host species as subscript, with the following abbreviations used: mus–moustached monkey (*Cercopithecus cephus*), mnd–mandrill (*Mandrillus sphinx*), rcm–red-capped mangabey (*Cercocebus torquatus*), sm–sooty mangabey (*Cercocebus atys*), deb–De Brazza's monkey (*Cercopithecus neglectus*), syk–Syke's monkey (*Cercopithecus albogularis*), agm–african green monkey (*Chlorocebus* spec).

Vpr$_{HIV-1}$ is important for virus replication *in vivo* and in macrophage infection models [26]. Recent proteomic analyses revealed that DCAF1 specificity modulation by Vpr$_{HIV-1}$ proteins results in down-regulation of hundreds of host proteins in a DCAF1- and proteasome-dependent manner [27], including the previously reported Vpr$_{HIV-1}$ degradation targets UNG2 [28], HLTF [29], MUS81 [30,31], MCM10 [32] and TET2 [33]. This surprising promiscuity in degradation targets is also partially conserved in more distant clades exemplified by Vpr$_{agm}$ and Vpr$_{mus}$ [27]. However, Vpr pleiotropy, and the lack of easily accessible experimental models, have prevented a characterisation of how these degradation events precisely promote replication [26].

By contrast, Vpx, exhibits a much narrower substrate range. It has recently been reported to target stimulator of interferon genes (STING) and components of the human silencing hub (HUSH) complex for degradation, leading to inhibition of antiviral cGAS-STING-mediated signalling and reactivation of latent proviruses, respectively [34–36]. Importantly, Vpx also recruits the SAMHD1 restriction factor to DCAF1, in order to mark it for proteasomal destruction [37,38]. SAMHD1 is a deoxynucleotide triphosphate (dNTP) triphosphohydrolase that restricts retroviral replication in non-dividing cells by lowering the dNTP pool to levels that cannot sustain viral reverse transcription [39–46]. Retroviruses that express Vpx are able to alleviate SAMHD1 restriction, allowing for replication in differentiated myeloid lineage cells, resting T cells and memory T cells [38,47,48]. As a result of the constant evolutionary arms race between the host's SAMHD1 restriction and its viral antagonist Vpx, the mechanism of Vpx-mediated SAMHD1 recruitment is highly virus species- and strain-specific: The Vpx clade represented by Vpx$_{HIV-2}$ recognises the SAMHD1 C-terminal domain (CtD), while Vpx$_{mnd2/rcm}$ binds the SAMHD1 N-terminal domain (NtD) in a fundamentally different way [24,49–52].

In the course of evolutionary adaptation to their primate hosts, due to selective pressure to evade SAMHD1 restriction, two groups of SIVs, SIV$_{agm}$ and SIV$_{deb/mus/syk}$, branched off from a common ancestor containing a Vpr protein which was unable to interact with SAMHD1, and *neo*-functionalised Vpr to bind SAMHD1 and induce its degradation. Subsequently, through a gene duplication or a recombination event, SIV and HIV clades exemplified by SIV$_{rcm}$ and HIV-2 gained the Vpx protein which took over the SAMHD1-degradation functionality. These viruses additionally encode for a Vpr protein with similar characteristics to the ancestral Vpr [24,49,53]. Consequently, SIV$_{agm}$ and SIV$_{deb/mus/syk}$ evolved "hybrid" Vpr proteins that retain targeting of some host factors depleted by HIV-1-type Vpr [27], and additionally induce SAMHD1 degradation.

To uncover the molecular mechanisms of DCAF1- and SAMHD1-interaction of such a "hybrid" Vpr, we initiated integrative biochemical and structural analyses of the Vpr protein from an SIV infecting *Cercopithecus cephus*, Vpr$_{mus}$. These studies reveal similarities and differences to Vpx and Vpr proteins from other lentivirus species and pinpoint the divergent molecular mechanism of Vpr$_{mus}$-dependent SAMHD1 recruitment to CUL4/ROC1/DDB1/DCAF1 (CRL4$^{DCAF1}$). Furthermore, cryo-electron microscopic (cryo-EM) reconstructions of a Vpr$_{mus}$-modified CRL4$^{DCAF1}$ protein complex allow for insights into the structural plasticity of the entire CRL4 ubiquitin ligase assembly, with implications for the ubiquitin transfer mechanism.

## Results

### SAMHD1-CtD is necessary and sufficient for Vpr$_{mus}$-binding and ubiquitylation in vitro

To investigate the molecular interactions between Vpr$_{mus}$, the *neo*-substrate SAMHD1 from rhesus macaque and CRL4 subunits DDB1/DCAF1 C-terminal domain (DCAF1-CtD),

protein complexes were reconstituted *in vitro* from purified components and analysed by gel filtration (GF) chromatography. The different protein constructs that were employed are shown schematically in S1A Fig. In the absence of additional binding partners, Vpr$_{mus}$ is insoluble after removal of the GST affinity purification tag (S1B Fig) and accordingly could not be applied to the GF column. No interaction of SAMHD1 with DDB1/DCAF1-CtD could be detected in the absence of Vpr$_{mus}$ (S1C Fig). Analysis of binary protein combinations (Vpr$_{mus}$ and DDB1/DCAF1-CtD; Vpr$_{mus}$ and SAMHD1) shows that Vpr$_{mus}$ elutes together with DDB1/DCAF1-CtD (S1D Fig) or with SAMHD1 (S1E Fig). Incubation of Vpr$_{mus}$ with DDB1/DCAF1B and SAMHD1 followed by GF resulted in co-elution of all three components (Fig 1A). Together, these results show that Vpr$_{mus}$ forms stable binary and ternary protein complexes with DDB1/DCAF1-CtD and/or SAMHD1 *in vitro*. Furthermore, incubation with any of these interaction partners apparently stabilises Vpr$_{mus}$ by alleviating its tendency for aggregation/insolubility.

Previous cell-based assays indicated that residues 583–626 of rhesus macaque SAMHD1 (SAMHD1-CtD) are necessary for Vpr$_{mus}$-induced proteasomal degradation [49]. To test this finding in our *in vitro* system, constructs containing SAMHD1-CtD fused to T4 lysozyme (T4L-SAMHD1-CtD), or containing only the N-terminal domains of SAMHD1, and lacking SAMHD1-CtD (SAMHD1-ΔCtD), were incubated with Vpr$_{mus}$ and DDB1/DCAF1-CtD, and complex formation was assessed by GF chromatography. Analysis of the resulting chromatograms by SDS-PAGE shows that SAMHD1-ΔCtD did not co-elute with DDB1/DCAF1-CtD/Vpr$_{mus}$ (Fig 1B). By contrast, T4L-SAMHD1-CtD accumulated in the same elution peak as DDB1/DCAF1-CtD and Vpr$_{mus}$ (Fig 1C). These results confirm that SAMHD1-CtD is necessary for stable association with DDB1/DCAF1-CtD/Vpr$_{mus}$ *in vitro*, and demonstrate that SAMHD1-CtD is sufficient for Vpr$_{mus}$-mediated recruitment of the T4L-SAMHD1-CtD fusion construct to DDB1/DCAF1-CtD.

To correlate these data with enzymatic activity, *in vitro* ubiquitylation assays were conducted by incubating SAMHD1, SAMHD1-ΔCtD or T4L-SAMHD1-CtD with purified CRL4$^{DCAF1-CtD}$, E1 (UBA1), E2 (UBCH5C), ubiquitin and ATP. Input proteins are shown in S2A Fig, and control reactions in S2B and S2C Fig. In the absence of Vpr$_{mus}$, no SAMHD1 ubiquitylation was observed (Figs 1D and S2D), while addition of Vpr$_{mus}$ resulted in robust SAMHD1 ubiquitylation, as demonstrated by an upward shift of SAMHD1 in the SDS PAGE analysis, induced by covalent modification with increasingly more ubiquitin molecules, leading to almost complete loss of the band corresponding to unmodified SAMHD1 after 15 min incubation (Figs 1E and S2E). In agreement with the analytical GF data, SAMHD1-ΔCtD was not ubiquitylated in the presence of Vpr$_{mus}$ (Figs 1F and S2F). By contrast, T4L-SAMHD1-CtD was efficiently ubiquitylated, resulting in >90% loss of the band corresponding to unmodified T4L-SAMHD1-CtD after 15 min (Figs 1G and S2F). Again, these data substantiate the functional importance of SAMHD1-CtD for Vpr$_{mus}$-mediated recruitment to the CRL4$^{DCAF1}$ ubiquitin ligase.

## Crystal Structure analysis of apo- and Vpr$_{mus}$-bound DDB1/DCAF1-CtD protein complexes

To obtain structural information regarding Vpr$_{mus}$ and its mode of binding to the CRL4 substrate receptor DCAF1, the X-ray crystal structures of a DDB1/DCAF1-CtD complex, and DDB1/DCAF1-CtD/T4L-Vpr$_{mus}$ (residues 1–92) fusion protein ternary complex were determined. The structures were solved using molecular replacement and refined to resolutions of 3.1 Å and 2.5 Å respectively (S1 Table). Vpr$_{mus}$ adopts a three-helix bundle fold, stabilised by coordination of a zinc ion by His and Cys residues on Helix-1 and at the C-terminus (Fig 2A).

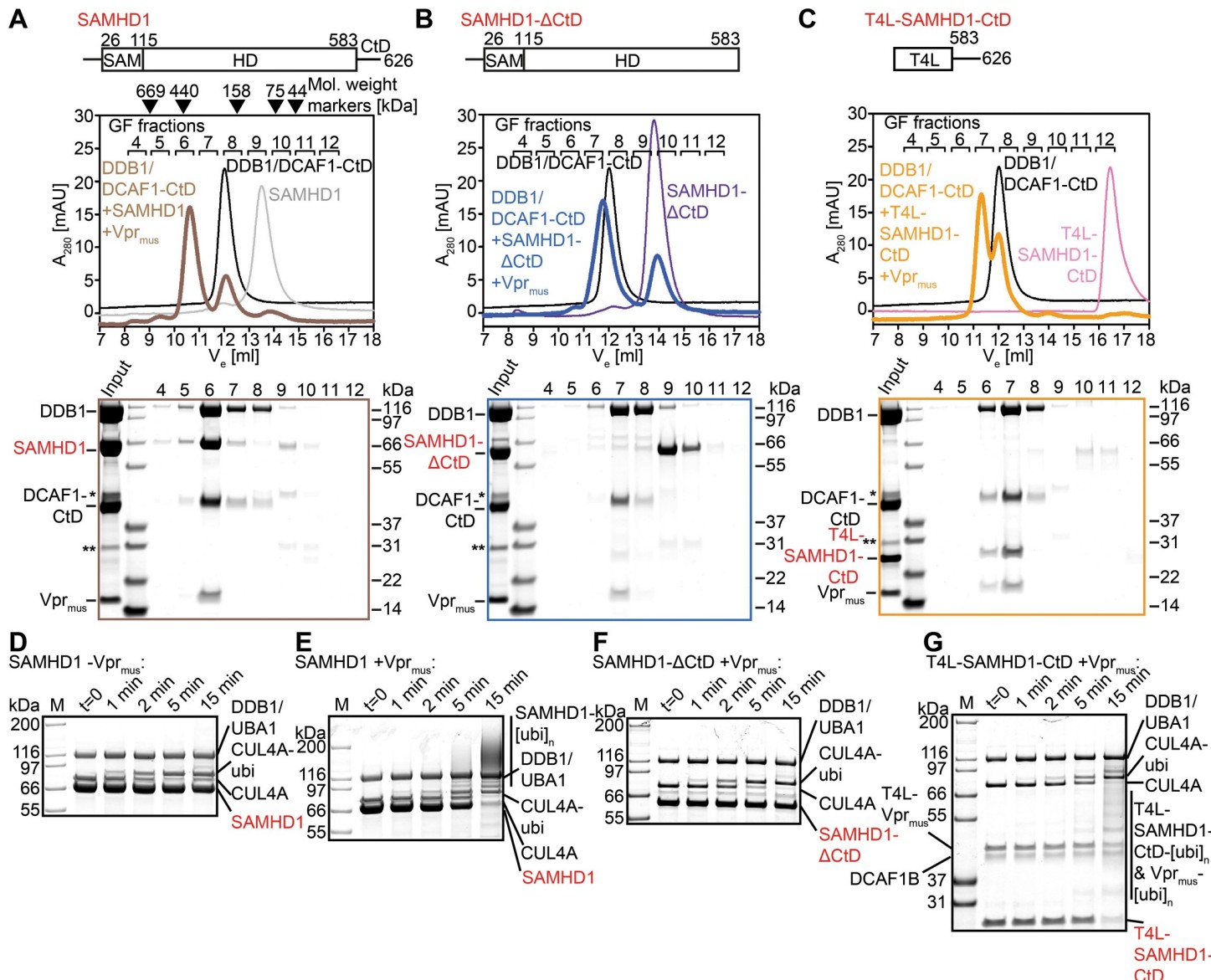

**Fig 1. Biochemical analysis of Vpr$_{mus}$-induced CRL4$^{DCAF1}$ specificity redirection.** (A-C) GF analysis of *in vitro* reconstitution of protein complexes containing DDB1/DCAF1-CtD, Vpr$_{mus}$ and SAMHD1 (**A**), SAMHD1-ΔCtD (**B**) or T4L-SAMHD1-CtD (**C**). Elution volumes of protein molecular weight standards are indicated above the chromatogram in **A**. Coomassie blue-stained SDS-PAGE analyses of fractions collected during the GF runs are shown below the chromatograms, with boxes colour-coded with respect to the chromatograms. SAM–sterile α-motif domain, HD–histidine-aspartate domain, T4L –T4 Lysozyme. The asterisk and double asterisk indicate slight contaminations with remaining GST-3C protease and the GST purification tag, respectively. (**D-G**) *In vitro* ubiquitylation reactions with purified protein components in the absence (**D**) or presence (**E-G**) of Vpr$_{mus}$, with the indicated SAMHD1 constructs as substrate. Reactions were stopped after the indicated times, separated on SDS-PAGE and visualised by Coomassie blue staining.

Superposition of Vpr$_{mus}$ with previously determined Vpx$_{sm}$ [50], Vpx$_{mnd2}$ [51,52], and Vpr$_{HIV-1}$ [54] structures reveals a conserved three-helix bundle fold, and similar position of the helix bundles on DCAF1-CtD (S3A Fig). In addition, the majority of side chains involved in DCAF1-interaction are type-conserved in all Vpx and Vpr proteins (S3B and S3C and S3D and S3E and S3F and S3G and S6A Figs), strongly suggesting a common molecular mechanism of host CRL4-DCAF1 hijacking by the Vpx/Vpr family of accessory proteins. However, there are also significant differences in helix length and register as well as conformational variation

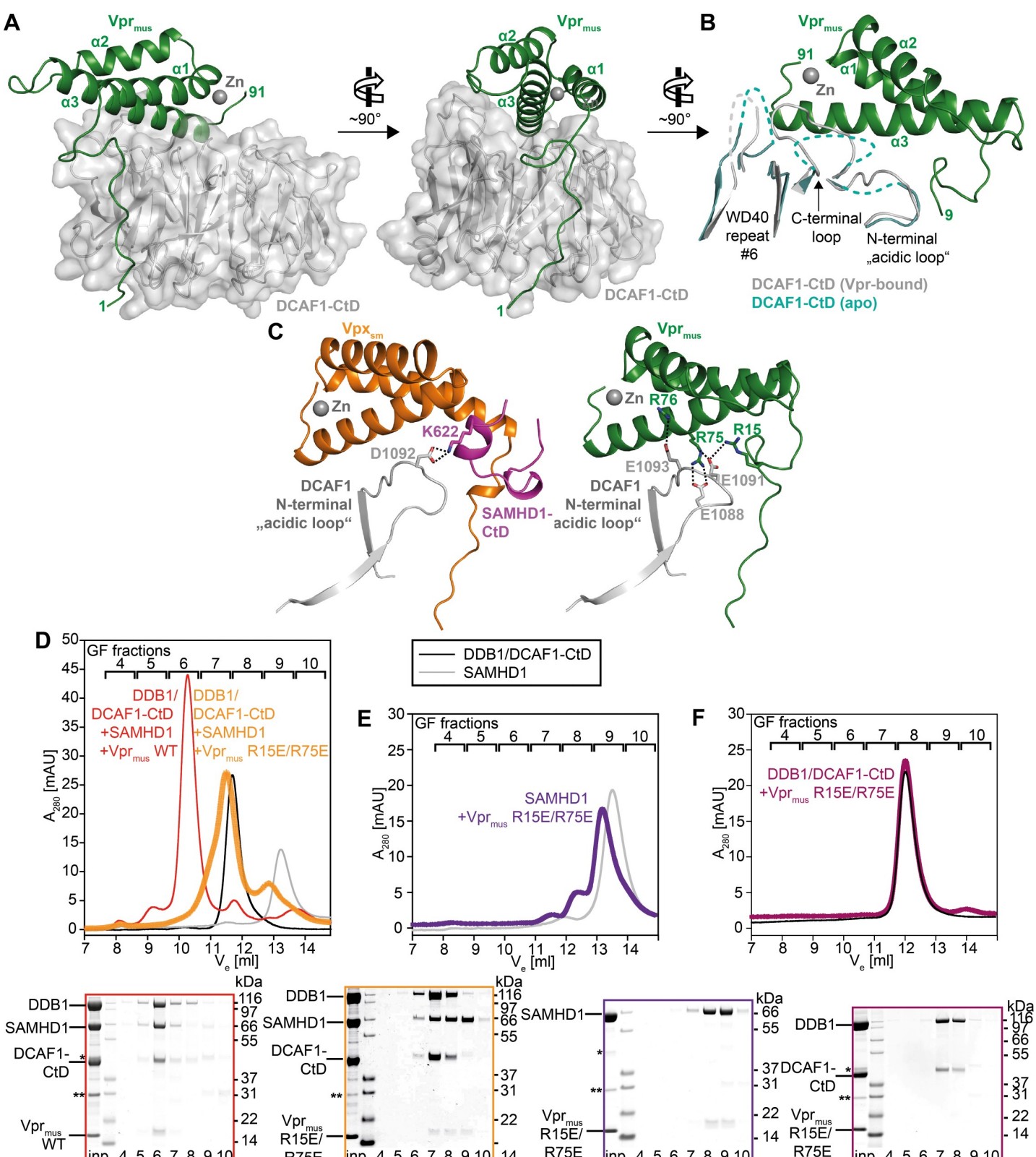

**Fig 2. Crystal structure of the DDB1/DCAF1-CtD/Vpr_mus complex.** (**A**) Overall structure of the DDB1/DCAF1-CtD/Vpr_mus complex in two views. DCAF1-CtD is shown as grey cartoon and semi-transparent surface. Vpr_mus is shown as a dark green cartoon with the co-ordinated zinc ion shown as grey sphere. T4L and DDB1 have been omitted for clarity. (**B**) Superposition of apo-DCAF1-CtD (light blue cartoon) with Vpr_mus-bound DCAF1-CtD (grey/green cartoon). Only DCAF1-CtD regions

with significant structural differences between apo- and Vpr$_{mus}$-bound forms are shown. Disordered loops are indicated as dashed lines. (**C**) Comparison of the binary Vpr$_{mus}$/DCAF1-CtD and ternary Vpx$_{sm}$/DCAF1-CtD/SAMHD1-CtD complexes. For DCAF1-CtD, only the N-terminal "acidic loop" region is shown. Vpr$_{mus}$, DCAF1-CtD and bound zinc are coloured as in **A**; Vpx$_{sm}$ is represented as orange cartoon and SAMHD1-CtD as pink cartoon. Selected Vpr/Vpx/DCAF1-CtD side chains are shown as sticks, and electrostatic interactions between these side chains are indicated as dotted lines. (**D**) GF analysis of *in vitro* reconstitution of protein complexes containing DDB1/DCAF1-CtD/Vpr$_{mus}$ or the Vpr$_{mus}$ R15E/R75E mutant, and SAMHD1. SDS-PAGE analyses of corresponding GF fractions are shown below the chromatogram, with boxes colour-coded with respect to the chromatogram. (**E-F**) *In vitro* reconstitution of protein complexes containing SAMHD1 and Vpr$_{mus}$ R15E/R75E (**E**) or DDB1/DCAF1-CtD and Vpr$_{mus}$ R15E/R75E (**F**). SDS-PAGE analyses of corresponding GF fractions are shown below the chromatogram, with boxes colour-coded with respect to the chromatogram. The asterisk and double asterisk indicate slight contaminations with remaining GST-3C protease and the GST purification tag, respectively.

in the loop region N-terminal of Helix-1, at the start of Helix-1 and in the loop between Helices-2 and -3 (S3A Fig).

Vpr$_{mus}$ binds to the side and on top of the disk-shaped 7-bladed β-propeller (BP) DCAF1-CtD domain with a total contact surface area of ~1600 Å$^2$ comprising three major regions of interaction. The extended Vpr$_{mus}$ N-terminus attaches to the cleft between DCAF1 BP blades 1 and 2 through several hydrogen bonds, electrostatic and hydrophobic interactions (S3B and S3C and S3D Fig). A second, smaller contact area is formed by hydrophobic interaction between Vpr$_{mus}$ residues L31 and E34 from Helix-1, and DCAF1 W1156, located in a loop on top of BP blade 2 (S3E Fig). The third interaction surface comprises the C-terminal half of Vpr$_{mus}$ Helix-3, which inserts into a ridge on top of DCAF1 (S3F and S3G Fig).

Superposition of the apo-DDB1/DCAF1-CtD and Vpr$_{mus}$-bound crystal structures reveals conformational changes in DCAF1 upon Vpr$_{mus}$ association. Binding of the N-terminal arm of Vpr$_{mus}$ induces only a minor rearrangement of a loop in BP blade 2 (S3C Fig). By contrast, significant structural changes occur on the upper surface of the BP domain: polar and hydrophobic interactions of DCAF1 residues P1329, F1330, F1355, N1371, L1378, M1380 and T1382 with Vpr$_{mus}$ side chains of T79, R83, R86 and E87 in Helix-3 result in the stabilisation of the sequence stretch that connect BP blades 6 and 7 ("C-terminal loop", Figs 2B and S3F). Moreover, side chain electrostatic interactions of Vpr$_{mus}$ residues R15, R75 and R76 with DCAF1 E1088, E1091 and E1093 lock the conformation of an "acidic loop" upstream of BP blade 1, which is also unstructured and flexible in the absence of Vpr$_{mus}$ (Figs 2B and 2C and S3D and S3E and S3F).

Notably, in previously determined structures of Vpx/DCAF1/SAMHD1 complexes the "acidic loop" is a central point of ternary contact, providing a binding platform for positively charged amino acid side chains in either the SAMHD1 N- or C-terminus [50–52]. For example, Vpx$_{sm}$ positions SAMHD1-CtD in such a way, that SAMHD1 K622 engages in electrostatic interaction with the DCAF1 "acidic loop" residue D1092 (Fig 2C, left panel). However, in the Vpr$_{mus}$ crystal structure the bound Vpr$_{mus}$ now blocks access to the corresponding SAMHD1-CtD binding pocket, in particular by the positioning of an extended N-terminal loop that precedes Helix-1. Additionally, Vpr$_{mus}$ side chains R15, R75 and R76 neutralise the DCAF1 "acidic loop", precluding the formation of further salt bridges to basic residues in SAMHD1-CtD (Fig 2C, right panel).

To validate the importance of Vpr$_{mus}$ residues R15 and R75 for DCAF1-CtD-binding, charge reversal mutations to glutamates were generated by site-directed mutagenesis. The circular dichroism (CD) spectrum of the Vpr$_{mus}$ R15E R75E double mutant GST-fusion protein was identical to the wild type, indicating similar secondary structure content and thus no major structural disturbances caused by the amino acid substitutions (S3H Fig). The effect of the Vpr$_{mus}$ R15E R75E double mutant on complex assembly was then analysed by GF chromatography. SDS-PAGE analysis of the resulting chromatographic profile shows an almost complete loss of the DDB1/DCAF1-CtD/Vpr$_{mus}$/SAMHD1 complex peak (Fig 2D, fraction 6), when compared to the wild type, concomitant with enrichment of (i) some proportion of

Vpr$_{mus}$ R15E R75E-bound DDB1/DCAF1-CtD (Fig 2D, fraction 7), (ii) free DDB1/DCAF1-CtD (fraction 7–8), and of (iii) Vpr$_{mus}$ R15E R75E/SAMHD1 binary complex (Fig 2D, fraction 8–9). This suggests that charge reversal of Vpr$_{mus}$ side chains R15 and R75 weakens the strong association with DCAF1 observed in wild type Vpr$_{mus}$, due to loss of electrostatic interaction with the "acidic loop", in accordance with the crystal structure. Consequently, some proportion of Vpr-bound SAMHD1 dissociates. This notion is further supported by GF analysis of binary combinations of the Vpr$_{mus}$ R15E R75E double mutant with either SAMHD1 or DDB1/DCAF1-CtD. Incubation of Vpr$_{mus}$ R15E R75E with SAMHD1 followed by GF leads to co-elution of both proteins, concomitant with a shift of the elution peak towards higher apparent molecular weight, compared to SAMHD1 alone (Fig 2E, fractions 8–9). By contrast, incubation of the Vpr$_{mus}$ double mutant with DDB1/DCAF1-CtD does not change the elution volume of the DDB/DCAF1-CtD species, and no band corresponding to Vpr can be detected in the SDS-PAGE analysis of the corresponding fractions (Fig 2F, fractions 7–8). These data clearly demonstrate loss of interaction with DDB1/DCAF1-CtD upon charge reversal of Vpr$_{mus}$ residues 15 and 75, while the SAMHD1-binding activity is retained.

## Cryo-EM analysis of CRL4-NEDD8$^{DCAF1-CtD}$/Vpr$_{mus}$/SAMHD1 conformational states and dynamics

To obtain structural insight into Vpr$_{mus}$ in the context of a complete CRL4 assembly, and to understand the SAMHD1 recruitment mechanism, we initiated cryo-EM analyses of the CRL4$^{DCAF1-CtD}$/Vpr$_{mus}$/SAMHD1 assembly. In these studies, the small ubiquitin-like protein NEDD8 was enzymatically attached to the CUL4 subunit, in order to obtain its active form (S4A Fig) [55]. A CRL4-NEDD8$^{DCAF1-CtD}$/Vpr$_{mus}$/SAMHD1 complex was reconstituted *in vitro* and purified by GF chromatography (S4B Fig). 2D classification of the resulting particle images revealed considerable compositional and conformational heterogeneity, especially regarding the presence and position of the CUL4-NEDD8/ROC1 sub-complex (stalk) relative to DDB1/DCAF1/Vpr$_{mus}$ (core) (S4C Fig).

Two consecutive rounds of 3D classification yielded three particle populations, resulting in 3D reconstructions at 8–10 Å resolution, which contained both the Vpr$_{mus}$-bound CRL4 core and the stalk (conformational states-1, -2 and -3, Figs 3A and S4D and S4E and S4F). The quality of the 3D volumes was sufficient to fit crystallographic models of core (Fig 2) and the stalk (PDB 2hye) [15] as rigid bodies (Figs 3B and S4G). For the catalytic RING-domain subunit ROC1, only fragmented electron density was present near the position it occupies in the crystallographic model (S4G Fig). In all three states, electron density was selectively absent for the C-terminal CUL4 winged helix B (WHB) domain (residues 674–759), which contains the NEDD8 modification site (K705), and for the preceding α-helix, which connects the CUL4 N-terminal domain to the WHB domain (S4G Fig). In accordance with this observation, the positions of CRL5-attached NEDD8 and of the CRL4 ROC1 RING domain are sterically incompatible upon superposition of their respective crystal structures (S4H Fig) [56].

Alignment of 3D volumes from states-1, -2 and -3 shows that core densities representing DDB1 BPA, BPC, DCAF1-CtD and Vpr$_{mus}$ superimpose well, indicating that these components do not undergo major conformational fluctuations and thus form a rigid platform for substrate binding and attachment of the CRL4 stalk (Fig 3). However, rotation of DDB1 BPB around a hinge connecting it to BPC results in three different orientations of state-1, -2 and -3 stalk regions relative to the core. BPB rotation angles were measured as 69˚ between state-1 and -2, and 50˚ between state-2 and -3.

These data are in line with previous prediction based on extensive comparative crystal structure analyses, which postulated an approx. 150˚ rotation of the CRL4 stalk around the

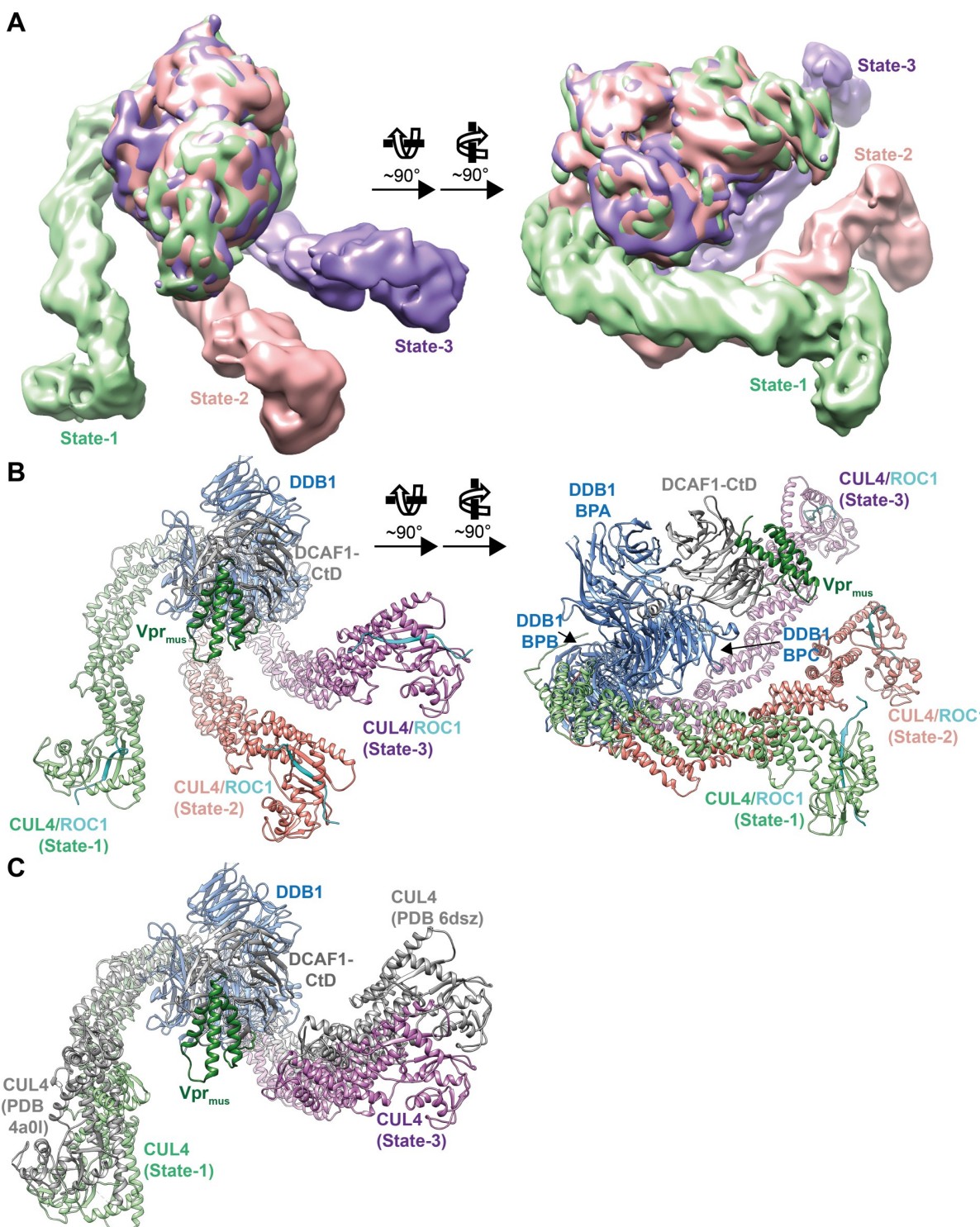

**Fig 3. Cryo-EM analysis of CRL4-NEDD8$^{\text{DCAF1-CtD}}$/Vpr$_{\text{mus}}$/SAMHD1 conformational states.** (**A**) Two views of an overlay of CRL4-NEDD8$^{\text{DCAF1-CtD}}$/Vpr$_{\text{mus}}$/SAMHD1 cryo-EM reconstructions (conformational state-1 –light green, state-2 –salmon, state-3 –purple). The portions of the densities corresponding to DDB1 BPA/BPC, DCAF1-CtD and Vpr$_{\text{mus}}$ have been superimposed. (**B**) Two views of a superposition of DDB1/DCAF1-CtD/Vpr$_{\text{mus}}$ and CUL4/ROC1 (PDB 2hye) [15] molecular models, which have been fitted as rigid bodies to the corresponding cryo-EM densities; the models are oriented as in **A**. DDB1/DCAF1-CtD/Vpr$_{\text{mus}}$ is shown as in Fig 2A, CUL4 is shown as cartoon, coloured as in **A** and ROC1 is shown as cyan cartoon. (**C**) Comparison of outermost CUL4 stalk orientations observed in the cryo-EM analysis presented here (states-1 and -3, coloured as in **B**, show 119.5˚ rotation of DDB1 BPB) to the two most extreme stalk positions present in previous crystal structures (PDB 4a0l [13], PDB 6dsz [123], coloured grey, show 143.4˚ DDB1 BPB rotation).

core [13,15,16,19,57]. However, the left- and rightmost CUL4 orientations observed here, states-1 and -3 from our cryo-EM analysis, indicate a slightly narrower stalk rotation range (119˚), when compared to the outermost stalk conformations modelled from previously determined crystal structures (143˚) (Fig 3C). A possible explanation for this discrepancy arises from inspection of the cryo-EM densities and fitted models, revealing that along with the main interaction interface on DDB1 BPB there are additional molecular contacts between CUL4 and DDB1. Specifically, in state-1, there is a contact between the loop connecting helices D and E of CUL4 cullin repeat (CR)1 (residues 161–169) and a loop protruding from BP blade 3 of DDB1 BPC (residues 795–801, S4I Fig). In state-3, the loop between CUL4 CR2 helices D and E (residues 275–282) abuts a region in the C-terminal helical domain of DDB1 (residues 1110–1127, S4J Fig). These auxiliary interactions might be required to lock the outermost stalk positions observed here in order to confine the rotation range of CUL4.

## Molecular mechanism of SAMHD1-targeting

A reanalysis of the cryo-EM data involving template-based particle picking and extensive 3D classification allowed for separation of an additional homogeneous particle population (S5A and S5B Fig). This subset of particle images yielded a 3D reconstruction at a nominal resolution of 7.3 Å that only contained electron density corresponding to the CRL4 core (S5A and S5B and S5C and S5D Fig). Molecular models of DDB1 BP domains A and C (BPA, BPC), DCAF1-CtD and Vpr$_{mus}$, derived from our crystal structure (Fig 2), could be fitted as rigid bodies into this cryo-EM volume (Fig 4A). No obvious electron density was visible for the bulk of SAMHD1. However, close inspection revealed an additional tubular, slightly arcing density feature, approx. 35 Å in length, located on the upper surface of the Vpr$_{mus}$ helix bundle, approximately 17 Å away from and opposite of the Vpr$_{mus}$/DCAF1-CtD binding interface (Fig 4A, red arrows). One end of the tubular volume contacts the middle of Vpr$_{mus}$ Helix-1, and the other end forms additional contacts to the C-terminus of Helix-2 and the N-terminus of Helix-3 (Fig 4B). A local resolution of 7.5–8 Å (S5C Fig) precluded the fitting of an atomic model. Considering the biochemical data, showing that SAMHD1-CtD is sufficient for recruitment to DDB1/DCAF1/Vpr$_{mus}$, we hypothesise that this observed electron density feature corresponds to a region of SAMHD1-CtD which physically interacts with Vpr$_{mus}$. Given its dimensions, the putative SAMHD1-CtD density could accommodate approx. 10 amino acid residues in a fully extended conformation or up to 23 residues in a kinked helical arrangement. All previous crystal structure analyses [46], as well as secondary structure predictions indicate that SAMHD1 residues C-terminal to the catalytic HD domain and C-terminal lobe (amino acids 599–626) are disordered in the absence of additional binding partners. Accordingly, the N-terminal globular domains of the SAMHD1 molecule might be flexibly linked to the C-terminal tether identified here. In that case, the bulk of SAMHD1 samples a multitude of positions relative to the DDB1/DCAF1-CtD/Vpr$_{mus}$ core, and consequently is averaged out in the process of cryo-EM reconstruction.

To test this hypothesis, Vpr$_{mus}$ amino acid residues in close proximity to the putative SAMHD1-CtD density were substituted by site-directed mutagenesis. Specifically, Vpr$_{mus}$ W29 was changed to alanine to block a hydrophobic contact with SAMHD1-CtD involving the aromatic side chain, and Vpr$_{mus}$ A66 was changed to a bulky tryptophan, in order to introduce a steric clash with SAMHD1-CtD (Fig 4B). The structural integrity of the Vpr$_{mus}$ W29A A66W double mutant was confirmed by CD spectroscopy (S3H Fig), and it was then assessed for complex formation with DDB1/DCAF1-CtD and SAMHD1 by analytical GF. In comparison to wild type Vpr$_{mus}$, the W29A A66W mutant showed reduced DDB1/DCAF1-CtD/Vpr$_{mus}$/SAMHD1 complex peak intensity (Fig 4C, fraction 6), concomitant with (i)

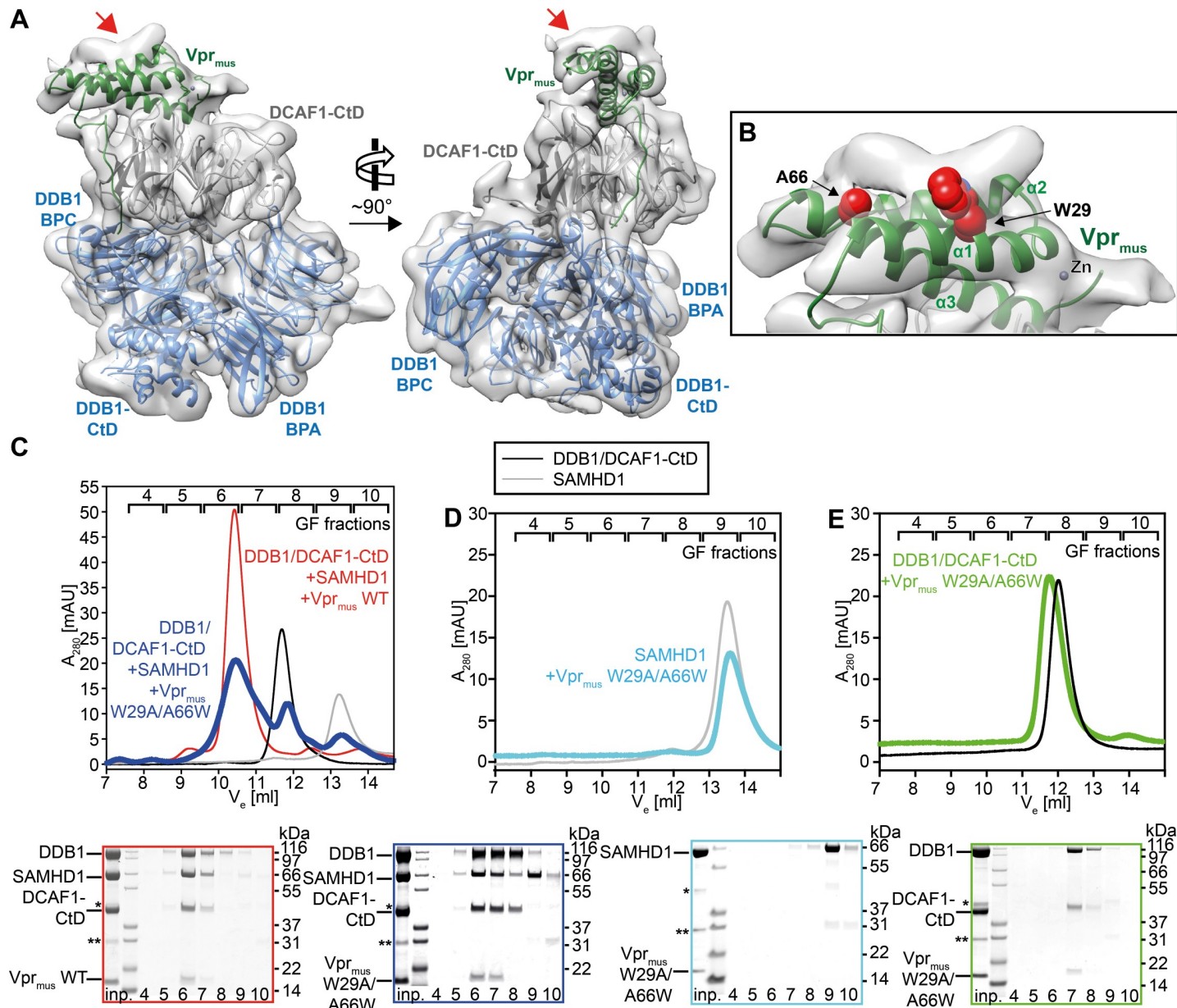

**Fig 4. Mechanism of SAMHD1-CtD recruitment by Vpr_mus.** (**A**) Two views of the cryo-EM reconstruction of the CRL4-NEDD8$^{DCAF1-CtD}$/Vpr_mus/SAMHD1 core. The crystal structure of the DDB1/DCAF1-CtD/Vpr_mus complex was fitted as a rigid body into the cryo-EM density and is shown in the same colours as in Fig 2A. The DDB1 BPB model and density was removed for clarity. The red arrows mark additional density on the upper surface of the Vpr_mus helix bundle. (**B**) Detailed view of the SAMHD1-CtD electron density. The model is in the same orientation as in **A**, left panel. Selected Vpr_mus residues W29 and A66, which are in close contact to the additional density, are shown as red space-fill representation. (**C**) *In vitro* reconstitution of protein complexes containing DDB1/DCAF1-CtD, Vpr_mus or the Vpr_mus W29A/A66W mutant, and SAMHD1, assessed by analytical GF. SDS-PAGE analyses of corresponding GF fractions are shown below the chromatogram, with boxes colour-coded with respect to the chromatogram. (**D-E**) *In vitro* reconstitution of protein complexes containing SAMHD1 and Vpr_mus W29A/A66W (**D**) or DDB1/DCAF1-CtD and Vpr_mus W29A/A66W (**E**). SDS-PAGE analyses of corresponding GF fractions are shown below the chromatogram, with boxes colour-coded with respect to the chromatogram. The asterisk and double asterisk indicate slight contaminations with remaining GST-3C protease and the GST purification tag, respectively.

enrichment of DDB1/DCAF1-CtD/Vpr_mus ternary complex, sub-stoichiometrically bound to SAMHD1 (Fig 4C, fraction 7), (ii) excess DDB1/DCAF1-CtD complex (Fig 4C, fraction 8), and (iii) excess SAMHD1 (Fig 4C, fractions 9–10). In addition, binary combinations of the Vpr_mus W29A A66W double mutant with DDB1/DCAF1-CtD or SAMHD1 were also

analysed by GF. These data show loss of SAMHD1 interaction (Fig 4D), while the ability to bind DDB1/DCAF1-CtD is retained (Fig 4E). Together, these biochemical analyses support a location of the SAMHD1-CtD binding site on the upper surface of the $Vpr_{mus}$ helix bundle, as suggested by medium-resolution cryo-EM reconstruction.

To obtain additional experimental evidence, the $CRL4^{DCAF1-CtD}/Vpr_{mus}/SAMHD1$ assembly was further examined by cross-linking mass spectrometry (CLMS), using the photo-reactive cross-linker sulfo-SDA. This bi-functional compound contains an NHS-ester functional group on one end that reacts with primary amines and hydroxyl groups, while the other end covalently links to any amino acid sidechain within reach upon UV-activation via a carbene intermediate [58]. Accordingly, incubation of proteins or protein complexes with sulfo-SDA, followed by UV-illumination, allows for high-density cross-linking of lysine, and to a lesser extent serine, threonine and tyrosine side chains to amino acids within reach of the SDA spacer group, with faster kinetics than pure NHS ester-based cross-linkers, due to the short half-life of the UV-activated intermediate. Cross-linked peptides are subsequently identified by mass spectrometry, and provide insights into the topology and residue-residue distances of proteins and complexes [59]. In the case of the $CRL4^{DCAF1-CtD}/Vpr_{mus}/SAMHD1$ assembly, the majority of identified cross-links that can be mapped onto the structure (468/519, 90.2%) are within the 25 Å violation threshold imposed by the geometry of the SDA spacer. Interestingly, 8 of the 11 cross-links between DDB1 and CUL4 are satisfied by the state-1 model, but increasingly violated in states-2 and -3 (S5E Fig), supporting in solution the rotational flexibility of the CRL4 stalk with respect to DDB1/DCAF1-CtD, as observed in cryo-EM (Fig 3).

An additional 300 cross-links involved SAMHD1, extending to the C-terminal half of CUL4, to a DDB1 sequence stretch comprising amino acid residues 900–1000, to parts of DCAF1-CtD and to $Vpr_{mus}$ (Figs 5A and S5E). The $CRL4^{DCAF1-CtD}/Vpr_{mus}$ residues exhibiting cross-links to SAMHD1 were mapped onto the state-2 model, and showed the presence of a large, yet defined, interaction surface (Fig 5B). Importantly, cross-links were apparent between the C-terminus of SAMHD1-CtD (residues K622, K626) and a region in $Vpr_{mus}$ Helix-1 (residues 27–36), which forms a part of the putative SAMHD1-CtD binding interface observed in cryo-EM, and which contains $Vpr_{mus}$ W29, one of the residues substituted in the mutagenesis and biochemical analysis presented above (Fig 5B, purple spheres). In addition, amino acid residues from the N-terminal portion of SAMHD1-CtD (residues K595, K596, T602-S606) cross-linked close to the DCAF1-CtD "acidic loop" (residues 1092–1096), which is immobilised by Vpr near the proposed SAMHD1-CtD binding site, and to the very C-terminus of CUL4 (residues Y744, A759), which is also adjacent to the predicted SAMHD1-CtD binding position (Fig 5B, pink spheres). Lastly, cross-links in the SAMHD1 N-terminal SAM and catalytic HD domains almost exclusively involved patches of the $CRL4^{DCAF1-CtD}/Vpr_{mus}$ surface surrounding and facing towards the putative SAMHD1-CtD attachment point (Fig 5B, light brown spheres). These observations are compatible with recruitment of SAMHD-CtD on the upper surface of the Vpr helix bundle, as indicated by cryo-EM. In addition, the spatial distribution of cross-links involving the SAMHD1 N-terminal domains suggest that these are flexibly connected to SAMHD1-CtD, leading to highly variable positioning relative to $CRL4^{DCAF1-CtD}/Vpr_{mus}$ and thus offering a multitude of cross-linking opportunities to nearby CRL4 components, again in line with the cryo-EM reconstruction results, especially upon consideration of the positional heterogeneity of the CUL4 stalk (Fig 3).

In order to evaluate the distance information inherent in SAMHD1-CtD cross-links in a more quantitative way, the volume accessible to SAMHD1-CtD for interaction with $CRL4^{DCAF1-CtD}/Vpr_{mus}$, consistent with the CLMS distance restraints, was simulated using the DisVis software tool [60,61]. For this analysis, SAMHD1-CtD was modelled as peptide in extended conformation. During the simulation, the state-2 $CRL4^{DCAF1-CtD}/Vpr_{mus}$ molecular

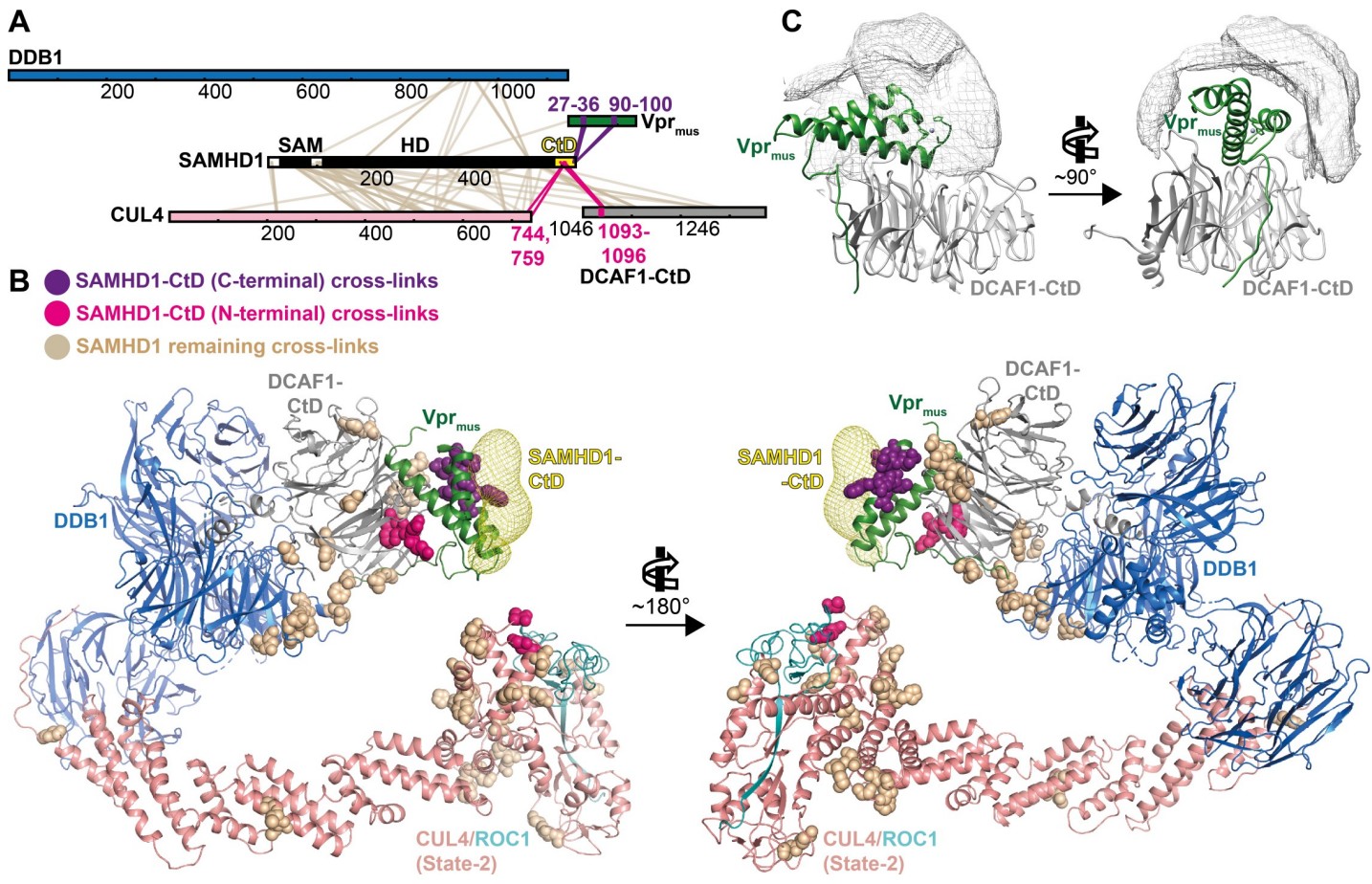

**Fig 5. Cross-linking mass spectrometry (CLMS) analysis of CRL4^{DCAF1-CtD}/ Vpr_{mus}/SAMHD1.** (**A**) Schematic representation of sulfo-SDA cross-links between CRL4^{DCAF1}/Vpr_{mus} and SAMHD1, identified by CLMS. Proteins are colour-coded as in Figs 3 and 4, and SAMHD1 black/white. SAMHD1-CtD is highlighted in yellow. Crosslinks to the N-terminal SAMHD1 globular SAM and HD domains are coloured light brown, while cross-links to the N-terminal half of SAMHD1-CtD are highlighted in pink and cross-links to the C-terminal end of SAMHD1-CtD are coloured purple. (**B**) Sulfo-SDA cross-links from **A**, in the same colour scheme, mapped on the molecular model of CRL4-NEDD8^{DCAF1-CtD}/Vpr_{mus}/SAMHD1 (state-2), obtained from cryo-EM analysis (Fig 3). SAMHD1-CtD density from the CRL4-NEDD8^{DCAF1-CtD}/Vpr_{mus}/SAMHD1 (core) cryo-EM analysis (Fig 4) is shown as yellow mesh. (**C**) The accessible interaction space of SAMHD1-CtD, calculated by the DisVis server [61], consistent with at least 14 of 26 observed cross-links, is visualised as grey mesh. DCAF1-CtD and Vpr_{mus} are oriented and coloured as in Fig 4A.

model was kept fixed, and a six-dimensional search of all possible degrees of freedom of rotation and translation for the SAMHD1-CtD model in molecular contact with CRL4^{DCAF1-CtD}/Vpr_{mus} was computed and ranked according to agreement with CLMS distance restraints. To visualise the output, all possible spatial positions of the centre of mass of SAMHD1-CtD, which satisfy >50% of the CLMS restraints, were plotted as density map on the structure of DCAF1-CtD/Vpr_{mus} (Fig 5C). In accordance with the cryo-EM reconstruction, this independent computational analysis also locates SAMHD1-CtD on top of the Vpr_{mus} helix bundle.

Taken together, the structural, biochemical and CLMS data are consistent with a model where the very C-terminus of SAMHD1 is recruited by Vpr_{mus}, to place the remaining SAMHD1 domains appropriately for access to the catalytic machinery at the distal end of the CRL4 stalk.

These data allow for structural comparison with *neo*-substrate binding modes of Vpx and Vpr proteins from different retrovirus lineages (Fig 6A and 6B and 6C and 6D). Vpx_{HIV-2} and Vpx_{sm} position SAMHD1-CtD at the side of the DCAF1 BP domain through interactions with

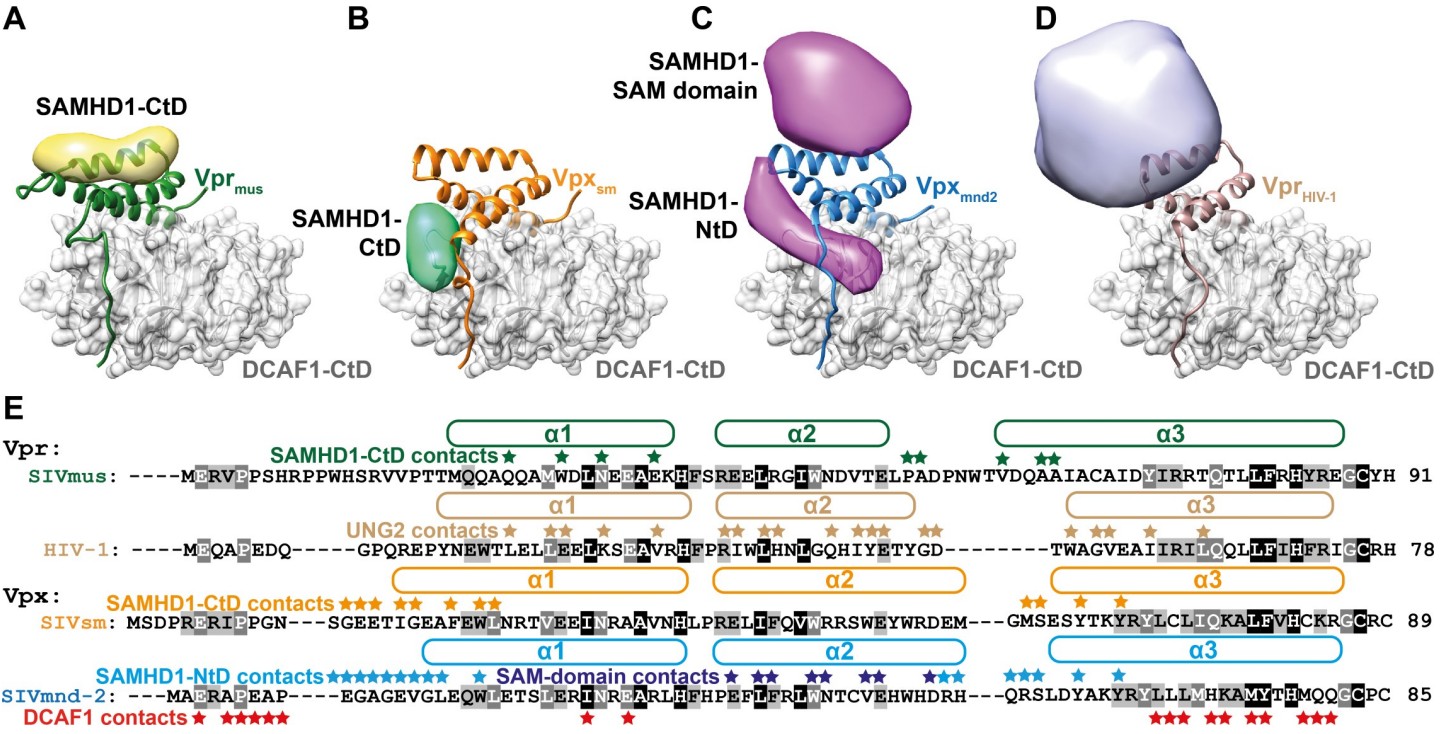

**Fig 6. Variability of *neo*-substrate recognition in Vpx/Vpr proteins.** Comparison of *neo*-substrate recognition modes of Vpr_mus (**A**), Vpx_sm (**B**), Vpx_mnd2 (**C**) and Vpr_HIV-1 (**D**). DCAF1-CtD is shown as grey cartoon and semi-transparent surface, Vpr_mus–green, Vpx_sm–orange, Vpx_mnd2–blue and Vpr_HIV-1– light brown are shown as cartoon. Models of the recruited ubiquitylation substrates are shown as strongly filtered, semi-transparent calculated electron density maps with the following colouring scheme: SAMHD1-CtD bound to Vpr_mus–yellow, SAMHD1-CtD (bound to Vpx_sm, PDB 4cc9) [50]–mint green, SAMHD1-NtD (Vpx_mnd2, PDB 5aja) [51]–magenta, UNG2 (Vpr_HIV-1, PDB 5jk7) [54]–light violet. (**E**) Multiple sequence alignment of Vpr and Vpx proteins from **A-D**. Helices are indicated by the boxes above the amino acid sequences. Residues involved in *neo*-substrate recognition are indicated by asterisks above the amino acid sequences. Residues involved in DCAF1-binding in all Vpr and Vpx proteins are indicated by red asterisks below the Vpr_mnd-2 amino acid sequence. Residues shaded grey or black are at least 60% or 90% type-conserved in all Vpx and Vpr proteins, respectively.

the N-termini of Vpx Helices-1 and -3 (Fig 6B) [50]. Vpx_mnd2 and Vpx_rcm bind SAMHD1-NtD using a bipartite interface comprising the side of the DCAF1 BP and the upper surface of the Vpx helix bundle (Fig 6C) [51,52]. Vpr_HIV-1 engages its ubiquitylation substrate UNG2 using both the top and the upper edge of the Vpr_HIV-1 helix bundle (Fig 6D) [54]. Of note, these upper-surface interaction interfaces only partially overlap with the Vpr_mus/SAMHD1-CtD binding interface identified here and employ fundamentally different sets of interacting amino acid residues (Figs 6E and S6A). Thus, it appears that the molecular interaction interfaces driving Vpx/Vpr-mediated *neo*-substrate recognition and degradation are not conserved between related SIV and HIV Vpx/Vpr accessory proteins, even in cases where identical SAMHD1-CtD regions are targeted for recruitment.

## Discussion

Our X-ray crystallographic studies of the DDB1/DCAF1-CtD/Vpr_mus assembly provide the first structural insight into a class of "hybrid" SIV Vpr proteins. These are present in the SIV_agm and SIV_mus/deb/syk lineages of lentiviruses and combine characteristics of related Vpr_HIV-1 and SIV Vpx accessory proteins.

Like SIV Vpx, "hybrid" Vpr proteins down-regulate the host restriction factor SAMHD1 by recruiting it to CRL4^DCAF1 for ubiquitylation and subsequent proteasomal degradation. However, using a combination of X-ray, cryo-EM and CLMS analyses, we show that the molecular

strategy, which $Vpr_{mus}$ evolved to target SAMHD1, is strikingly different from Vpx-containing SIV strains. In the two clades of Vpx proteins, divergent amino acid sequence stretches just upstream of Helix-1 (variable region (VR)1, S6A Fig), together with polymorphisms in the SAMHD1-N-terminus of the respective host species, determine if HIV-2-type or $SIV_{mnd}$-type Vpx recognise SAMHD1-CtD or SAMHD1-NtD, respectively. These recognition mechanisms result in positioning of SAMHD1-CtD or -NtD on the side of the DCAF1 BP domain in a way that allows for additional contacts between SAMHD1 and DCAF1, thus forming ternary Vpx/ SAMHD1/DCAF1 assemblies with very low dissociation rates [50–52,62]. In $Vpr_{mus}$, different principles determine the specificity for SAMHD1-CtD. Here, VR1 is not involved in SAMHD1-CtD-binding at all, but forms additional interactions with DCAF1, which are not observed in Vpx/DCAF1 protein complexes (S6A Fig). Molecular contacts between $Vpr_{mus}$ and SAMHD1 are dispersed on Helices-1 and -3, facing away from the DCAF1 interaction site and immobilising SAMHD1-CtD on the top side of the $Vpr_{mus}$ helix bundle (S6A Fig). Placement of SAMHD1-CtD in such a position precludes stabilising ternary interaction with DCAF1-CtD, but still results in robust SAMHD1 ubiquitylation *in vitro* and SAMHD1 degradation in cell-based assays [24]. Accordingly, our *in vitro* reconstitution analyses show that $Vpr_{mus}$ is able to form stable binary interaction with either SAMHD1 or DCAF1-CtD, in the absence of the respective third binding partner. This leaves the question unanswered, if in a physiological setting, upon host cell infection, $Vpr_{mus}$ first captures SAMHD1 to guide it to the CRL4 complex, or if it hijacks CRL4 to subsequently recruit SAMHD1. However, since CRL4 localises to both cytoplasm and nucleus [63], while SAMHD1 is exclusively found in the nucleus [64], it is tempting to speculate that upon entering the host cell, Vpr at first encounters and binds cytoplasmic CRL4$^{DCAF1}$, to subsequently translocate into the nucleus for SAMHD1 recruitment.

Predictions regarding the molecular mechanism of SAMHD1-binding by other "hybrid" Vpr orthologues are difficult due to sequence divergence. Even in $Vpr_{deb}$, the closest relative to $Vpr_{mus}$, only approximately 50% of amino acid side chains lining the putative SAMHD1-CtD binding pocket are conserved (S6A Fig). Previous *in vitro* ubiquitylation and cell-based degradation experiments did not show a clear preference of $Vpr_{deb}$ for recruitment of either SAMHD1-NtD or–CtD [24,49]. Furthermore, it is disputed if $Vpr_{deb}$ actually binds DCAF1 [65], which might possibly be explained by amino acid variations in the very N-terminus and/ or in Helix-3 (S6A Fig). $Vpr_{syk}$ is specific for SAMHD1-CtD [49], but the majority of residues forming the binding platform for SAMHD1-CtD observed in the present study are not conserved. The $SIV_{agm}$ lineage of Vpr proteins is even more divergent, with significant differences not only in possible SAMHD1-contacting residues, but also in the sequence stretches preceding Helix-1, and connecting Helices-2 and -3, as well as in the N-terminal half of Helix-3 (S6A Fig). Furthermore, there are indications that recruitment of SAMHD1 by the $Vpr_{agm.GRI}$ subtype involves molecular recognition of both SAMHD1-NtD and–CtD [49,53]. In conclusion, recurring rounds of evolutionary lentiviral adaptation to the host SAMHD1 restriction factor, followed by host re-adaptation, resulted in highly species-specific, diverse molecular modes of Vpr-SAMHD1 interaction. Similar molecular arms races between cell-intrinsic antiviral host factors and viral antagonists shaped the species-specific lentiviral antagonism of e.g. host restriction factors of the APOBEC3 family and tetherin, through induction of their degradation by the respective viral antagonists Vif or Nef/Vpu [66–68]. Furthermore, viral re-adaptation to certain simian and human variants of these restriction factors, following cross-species transmission, took part in the emergence of pandemic HIV strains, thus highlighting the importance of structural insight into these processes [9]. In addition to the instance presented here, further structural characterisation of SAMHD1-Vpr complexes will be necessary to fully define outcomes of this particular virus-host molecular arms race.

Previous structural investigation of DDB1/DCAF1/Vpr$_{HIV-1}$ in complex with the *neo*-substrate UNG2 demonstrated that Vpr$_{HIV-1}$ engages UNG2 by mimicking the DNA phosphate backbone. More precisely, UNG2 residues, which project into the major groove of its endogenous DNA substrate, insert into a hydrophobic cleft formed by Vpr$_{HIV-1}$ Helices-1, -2 and the N-terminal half of Helix-3 [54]. This mechanism might rationalise Vpr$_{HIV-1}$'s extraordinary binding promiscuity, since the list of potential Vpr$_{HIV-1}$ degradation substrates is significantly enriched in DNA- and RNA-binding proteins [27]. Moreover, promiscuous Vpr$_{HIV-1}$-induced degradation of host factors with DNA- or RNA-binding activity has been proposed to induce cell cycle arrest at the G2/M phase border, which is the most thoroughly described phenotype of Vpr proteins so far [26,27,69]. In Vpr$_{mus}$, the N-terminal half of Helix-1 as well as the bulky amino acid residue W48, which is also conserved in Vpr$_{agm}$ and Vpx, constrict the hydrophobic cleft (S6A and S6B Fig). Furthermore, the extended N-terminus of Vpr$_{mus}$ Helix-3 is not compatible with UNG2-binding due to steric exclusion (S6C Fig). In accordance with these observations, Vpr$_{mus}$ does not down-regulate UNG2 in a human T cell line [27]. However, Vpr$_{mus}$, Vpr$_{syk}$ and Vpr$_{agm}$ also cause G2/M cell cycle arrest in their respective host cells [65,70,71]. This strongly hints at the existence of further structural determinants in Vpr$_{mus}$, Vpr$_{syk}$, Vpr$_{agm}$ and potentially Vpr$_{HIV-1}$, which regulate recruitment and ubiquitylation of DNA/RNA-binding host factors, in addition to the hydrophobic, DNA-mimicking cleft on top of the three-helix bundle. Future efforts to structurally characterise these determinants will further extend our understanding of how the Vpx/Vpr helical scaffold binds, and in this way adapts to a multitude of *neo*-substrate epitopes.

Our cryo-EM reconstructions of CRL4$^{DCAF1-CtD}$/Vpr$_{mus}$/SAMHD1, complemented by CLMS, also provide insights into the structural dynamics of CRL4 assemblies prior to ubiquitin transfer. The data confirm previously described rotational movement of the CRL4 stalk, in the absence of constraints imposed by a crystal lattice, creating a ubiquitylation zone around the Vpr$_{mus}$-modified substrate receptor (Figs 3 and 7A) [13,15,16,19,57]. Missing density for the neddylated CUL4 WHB domain and for the catalytic ROC1 RING domain indicates that these distal stalk elements are highly mobile and likely sample a multitude of orientations relative to the CUL4 scaffold (S4G Fig). These observations are in line with structure analyses of CRL1 and CRL5, where CUL1/5 neddylation leads to re-orientation of the cullin WHB domain, and to release of the ROC1 RING domain from the cullin scaffold, concomitant with stimulation of ubiquitylation activity [56]. Moreover, recent cryo-EM structure analysis of CRL1$^{β-TRCP}$/IκBα demonstrated substantial mobility of pre-catalytic CUL1-NEDD8 WHB and ROC1 RING domains [72]. Such flexibility seems necessary to structurally organise multiple CRL1-dependent processes, in particular the nucleation of a catalytic assembly, involving intricate protein-protein interactions between NEDD8, CUL1, ubiquitin-charged E2 and substrate receptor. This synergistic assembly then steers the ubiquitin C-terminus towards a substrate lysine for priming with ubiquitin [72]. Accordingly, our cryo-EM studies might indicate that similar principles apply for CRL4-catalysed ubiquitylation. However, to unravel the catalytic architecture of CRL4, sophisticated cross-linking procedures as in reference [72] will have to be pursued.

The mobility of the CRL4 stalk might assist the accommodation of a variety of sizes and shapes of substrates in the CRL4 ubiquitylation zone and might rationalise the wide substrate range accessible to CRL4 ubiquitylation through multiple DCAF receptors (Fig 7A). Owing to selective pressure to counteract the host's SAMHD1 restriction, HIV-2 and certain SIVs have taken advantage of this dynamic CRL4 architecture by modification of the DCAF1 substrate receptor with Vpx/Vpr-family accessory proteins. In such a way, either SAMHD1-CtD or -NtD is tethered to DCAF1, in order to flexibly recruit the bulk of SAMHD1 to further improve the accessibility of lysine side chains both tether-proximal and on the SAMHD1 globular domains to the neddylated CRL4 catalytic assembly (Fig 7B and 7C).

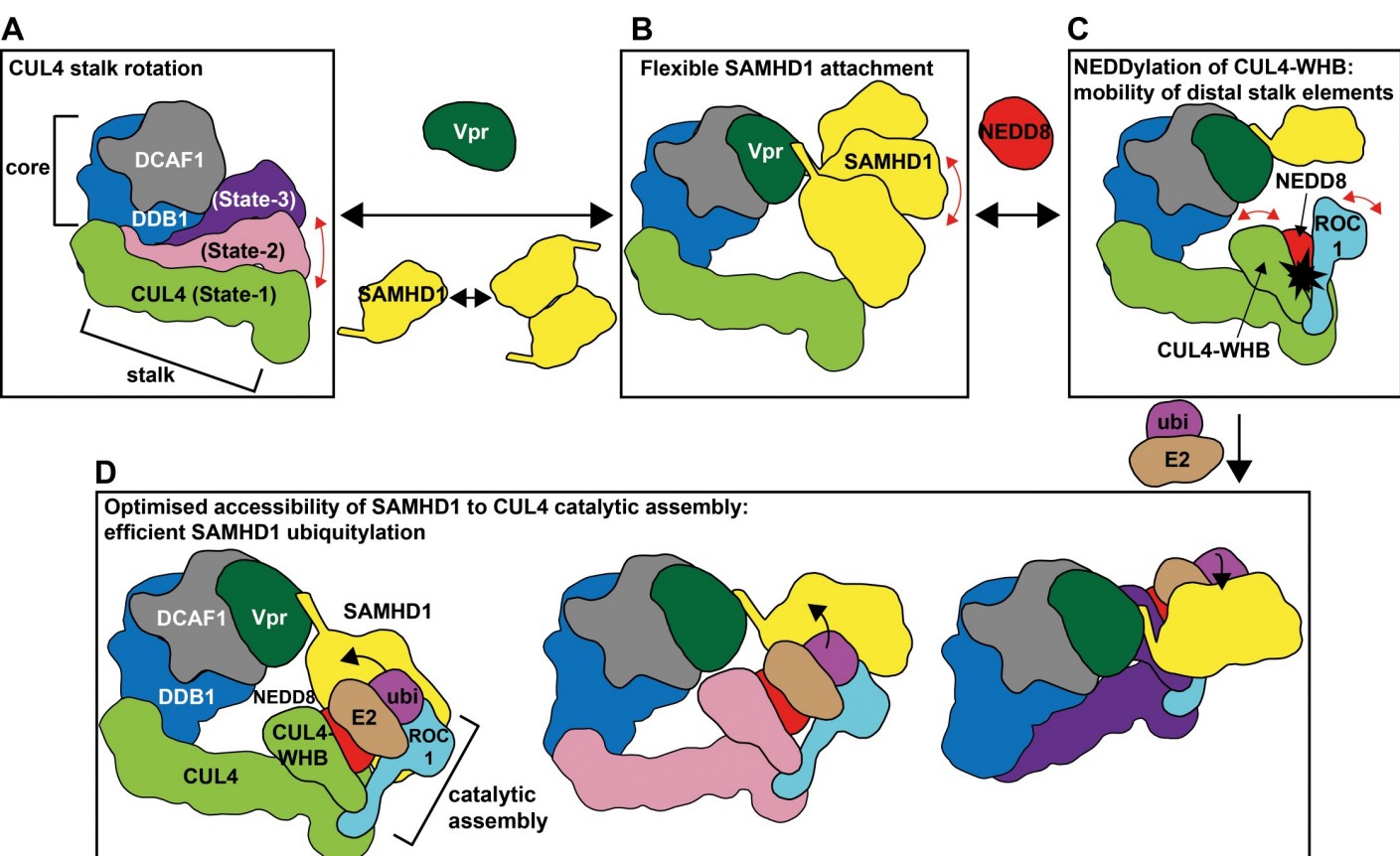

**Fig 7. Schematic illustration of structural plasticity in Vpr_mus-modified CRL4^DCAF1-CtD, and implications for ubiquitin transfer.** (**A**) Rotation of the CRL4 stalk increases the space accessible to catalytic elements at the distal tip of the stalk, forming a ubiquitylation zone around the core. (**B**) Flexible tethering of SAMHD1 to the core by Vpr_mus places the bulk of SAMHD1 in the ubiquitylation zone and optimises surface accessibility. Under the experimental conditions, SAMHD1 adopts a monomer-dimer equilibrium, with both forms being competent for Vpr-binding. In **B-C**, only monomeric SAMHD1 is schematically indicated for clarity. (**C**) Modification of CUL4-WHB with NEDD8, triggered by substrate binding, leads to increased mobility of these distal stalk elements (CUL4-WHB, ROC1 RING domain) [56], further extending the ubiquitylation zone and activating the formation of a catalytic assembly for ubiquitin transfer (see also **D**) [72]. (**D**) Dynamic processes **A-C** together create numerous possibilities for assembly of the catalytic machinery (CUL4-NEDD8 WHB, ROC1, ubiquitin-(ubi-)charged E2) on surface-exposed SAMHD1 lysine side chains. Here, three of these possibilities are exemplified schematically. In this way, ubiquitin coverage on SAMHD1 is maximised.

The catalytic dNTP triphosphohydrolase activity of SAMHD1 depends on nucleotide-dependent oligomerisation, mediated by two allosteric nucleotide-binding sites. In the absence of nucleotides, SAMHD1 adopts a monomer-dimer equilibrium with an equilibrium dissociation constant in the low micromolar range [73]. In the present work, SAMHD1 preparations and subsequent biochemical and structural studies have been performed without exogenously added nucleotides. Hence, under the experimental conditions, monomeric and dimeric states of SAMHD1 are expected to co-exist, competent for recruitment to Vpr. For clarity, only the binding of a monomeric SAMHD1 species is schematically indicated in Fig 7. However, recruitment of a SAMHD1 dimer might expose additional surface-exposed lysine residues to the CRL4 catalytic machinery and thus might further improve the efficacy of SAMHD1 ubiquitylation.

Insertion of guanine-based nucleotides in the first binding site shifts the SAMHD1 monomer-dimer equilibrium towards the dimeric form, and dNTP-binding to the second site leads to assembly of the catalytically active tetramer [41,73–78]. In accordance with the absence of such nucleotides, our analytical gel filtration data and cryo-EM reconstructions do not support

the existence of SAMHD1 tetramers under the experimental conditions (Figs 1 and S1 and S4 and S5). However, SAMHD1-CtD is essential for tetramer formation by contributing critical molecular contacts to neighbouring protomers [46]. Furthermore, tetramer destabilisation by CDK1/2-cyclinA-dependent phosphorylation of T592 in SAMHD1-CtD endogenously attenuates SAMHD1 activity in cycling cells [46,77,79,80]. Hence, under physiological conditions, it is conceivable that $Vpr_{mus}$ destabilises SAMHD1 tetramers by sequestering SAMHD1-CtD, in order to abrogate SAMHD1 activity, prior to inducing its proteasomal degradation. Such a mechanism would be in accordance with previous observation of $Vpx_{HIV-2}$-mediated SAMHD1 tetramer disassembly and inhibition of dNTPase activity [62].

Altogether, intrinsic CRL4 mobility, in combination with flexible Vpx/Vpr-mediated SAMHD1 recruitment maximises the efficiency of SAMHD1 poly-ubiquitylation and proteasomal degradation to stimulate virus replication (Fig 7D). In infected cells however, there is a stoichiometric mismatch between less than 1000 Vpr molecules, which are introduced in the host cell, and SAMHD1, which is abundant across a broad range of tissues and cell types [26,81]. Tight coupling of CRL4 to the p97 ATPase confers efficient unfolding of poly-ubiquitylated substrates prior to proteasomal degradation [82]. In this way, ubiquitylated SAMHD1 is removed from Vpr-bound $CRL4^{DCAF1}$ to initiate subsequent rounds of SAMHD1 recruitment, ubiquitylation and degradation.

Lastly, structural insight into this evolutionary optimised, highly specific protein degradation machinery might inform the positioning of novel $CRL4^{DCAF1}$-based synthetic modalities for targeted protein degradation, e.g. in the form of proteolysis-targeting chimera-(PROTAC-) or molecular glue-type compounds [83,84]. In addition, while current highly active antiretroviral therapy (HAART) regimens are able to control HIV-1 replication in infected patients [85], they cannot eradicate the virus due to viral rebound after treatment cessation, and they lead to emergence of resistant virus variants [86]. Accordingly, identification and inhibition of novel targets, in addition to those already covered by HAART, are of high interest. In this context, HIV accessory proteins have for a long time been regarded as promising drug targets [87,88]. For example, preclinical research aiming at inhibition of Vif-mediated APOBEC3 degradation culminated in the isolation of several compounds which impede HIV-1 replication [89–91]. Due to its promiscuous host interactions, $Vpr_{HIV-1}$ is also considered an attractive antiretroviral target [92]. The comparative structural analyses of Vpr/DCAF1-CtD interaction presented here might inform future efforts to disrupt this interaction interface with small molecules, in order to abolish all Vpr-induced host protein degradation processes at once, to maximise the inhibitory effect on virus replication.

## Methods

### Protein expression and purification

Constructs were PCR-amplified from cDNA templates and inserted into the indicated expression plasmids using standard restriction enzyme methods (S2 Table). pAcGHLT-B-DDB1 (plasmid #48638) and pET28-UBA1 (plasmid #32534) were obtained from Addgene. The pOPC-UBA3-GST-APPBP1 co-expression plasmid, and the pGex6P2-UBC12 plasmid were obtained from MRC-PPU Reagents and Services (clones 32498, 3879). Bovine erythrocyte ubiquitin and recombinant hsNEDD8 were purchased from Sigma-Aldrich (U6253) and Boston-Biochem (UL-812) respectively. Point mutations were introduced by site-directed mutagenesis using KOD polymerase (Novagen). All constructs and variants are summarised S3 Table.

Proteins expressed from vectors pAcGHLT-B, pGex6P1/2, pOPC and pET49b contained an N-terminal GST-His-tag; pHisSUMO–N-terminal His-SUMO-tag; pET28, pRSF-Duet-1 – N-terminal His-tag; pTri-Ex-6 –C-terminal His-tag. Constructs in vectors pAcGHLT-B and

pTri-Ex-6 were expressed in Sf9 cells, and constructs in vectors pET28, pET49b, pGex6P1/2, pRSF-Duet-1, and pHisSUMO in *E. coli* Rosetta 2(DE3).

Recombinant baculoviruses (*Autographa californica nucleopolyhedrovirus* clone C6) were generated as described previously [93]. Sf9 cells were cultured in Insect-XPRESS medium (Lonza) at 28°C in an Innova 42R incubator shaker (New Brunswick) at a shaking speed of 180 rpm. In a typical preparation, 1 L of Sf9 cells at $3\times10^6$ cells/mL were co-infected with 4 mL of high titre DDB1 virus and 4 mL of high titre DCAF1-CtD virus for 72 h.

For a typical *E. coli* Rosetta 2 (DE3) expression, 2 L of LB medium was inoculated with 20 mL of an overnight culture and grown in a Multitron HT incubator shaker (Infors) at 37°C, 150 rpm until $OD_{600}$ reached 0.7. At that point, temperature was reduced to 18°C, protein expression was induced by addition of 0.2 mM IPTG, and cultures were grown for further 20 h. During co-expression of CUL4 and ROC1 from pRSF-Duet, 50 µM zinc sulfate was added to the growth medium before induction.

Sf9 cells were pelleted by centrifugation at 1000 rpm, 4°C for 30 min using a JLA 9.1000 centrifuge rotor (Beckman). *E. coli* cells were pelleted by centrifugation at 4000 rpm, 4°C for 15 min using the same rotor. Cell pellets were resuspended in buffer containing 50 mM Tris, pH 7.8, 500 mM NaCl, 4 mM MgCl2, 0.5 mM tris-(2-carboxyethyl)-phosphine (TCEP), mini-complete protease inhibitors (1 tablet per 50 mL) and 20 mM imidazole (for His-tagged proteins only). 100 mL of lysis buffer was used for resuspension of a pellet from 1 L Sf9 culture, and 35 mL lysis buffer per pellet from 1 L *E. coli* culture. Before resuspension of CUL4/ROC1 co-expression pellets, the buffer pH was adjusted to 8.5. 5 µL Benzonase (Merck) was added and the cells lysed by passing the suspension at least twice through a Microfluidiser (Microfluidics). Lysates were clarified by centrifugation at 48000xg for 45 min at 4°C.

Protein purification was performed at 4°C on an Äkta pure FPLC (GE) using XK 16/20 chromatography columns (GE) containing 10 mL of the appropriate affinity resin. GST-tagged proteins were captured on glutathione-Sepharose (GSH-Sepharose FF, GE), washed with 250 mL of wash buffer (50 mM Tris-HCl pH 7.8, 500 mM NaCl, 4 mM MgCl$_2$, 0.5 mM TCEP), and eluted with the same buffer supplemented with 20 mM reduced glutathione. His-tagged proteins were immobilised on Ni-Sepharose HP (GE), washed with 250 mL of wash buffer supplemented with 20 mM imidazole, and eluted with wash buffer containing 0.3 M imidazole. Eluent fractions were analysed by SDS-PAGE, and appropriate fractions were pooled and reduced to 5 mL using centrifugal filter devices (Vivaspin). If applicable, 100 µg GST-3C protease, or 50 µg thrombin, per mg total protein, was added and the sample was incubated for 12 h on ice to cleave off affinity tags. As second purification step, gel filtration chromatography (GF) was performed on an Äkta prime plus FPLC (GE), with Superdex 200 16/600 columns (GE), equilibrated in 10 mM Tris-HCl pH 7.8, 150 mM NaCl, 4 mM MgCl$_2$, 0.5 mM TCEP buffer, at a flow rate of 1 mL/min. For purification of the CUL4/ROC1 complex, the pH of all purification buffers was adjusted to 8.5. Peak fractions were analysed by SDS-PAGE, appropriate fractions were pooled and concentrated to approx. 20 mg/mL, flash-frozen in liquid nitrogen in small aliquots and stored at -80°C. Protein concentrations were determined with a NanoDrop spectrophotometer (ND 1000, Peqlab), using theoretical absorption coefficients calculated based upon the amino acid sequence by ProtParam on the ExPASy webserver [94].

## Analytical gel filtration analysis

Prior to gel filtration analysis affinity tags were removed by incubation of 30 µg GST-3C protease with 6 µM of each protein component in a volume of 150 µL wash buffer, followed by incubation on ice for 12 h. In order to remove the cleaved GST-tag and GST-3C protease, 20 µL GSH-Sepharose FF beads (GE) were added and the sample was rotated at 4°C for one hour.

GSH-Sepharose beads were removed by centrifugation at 4˚C, 3500 rpm for 5 min, and 120 μL of the supernatant was loaded on an analytical GF column (Superdex 200 10/300 GL, GE), equilibrated in 10 mM Tris-HCl pH 7.8, 150 mM NaCl, 4 mM MgCl$_2$, 0.5 mM TCEP, at a flow rate of 0.5 mL/min. 1 mL fractions were collected and analysed by SDS-PAGE. For the following control samples, 120 μl of purified protein was applied directly to the GF column, because no purification tag had to be removed by cleavage: SAMHD1, T4L-SAMHD1-CtD, SAMHD1-ΔCtD. For these samples, the concentration was adjusted to 18 μM, 37 μM and 30 μM, respectively, to account for the lower extinction coefficient of these isolated protein components, in order to allow for better visualisation of the elution peak. The GF column was calibrated using the high molecular weight range (HMW) Gel Filtration Calibration Kit (GE) according to the manufacturer's instructions.

### *In vitro* ubiquitylation assays

160 μL reactions were prepared, containing 0.5 μM substrate (indicated SAMHD1 constructs, S2 Fig), 0.125 μM DDB1/DCAF1-CtD, 0.125 μM CUL4/ROC1, 0.125 μM HisSUMO-T4L-Vpr$_{mus}$ (residues 1–92), 0.25 μM UBCH5C, 15 μM ubiquitin in 20 mM Tris-HCl pH 7.8, 150 mM NaCl, 2.5 mM MgCl$_2$, 2.5 mM ATP. In control reactions, certain components were left out as indicated in S2 Fig. A 30 μl sample for SDS-PAGE analysis was taken (t = 0). Reactions were initiated by addition of 0.05 μM UBA1, incubated at 37˚C, and 30 μl SDS-PAGE samples were taken after 1 min, 2 min, 5 min and 15 min, immediately mixed with 10 μl 4x SDS sample buffer and boiled at 95˚C for 5 min. Samples were analysed by SDS-PAGE.

### *In vitro* neddylation of CUL4/ROC1

For initial neddylation tests, a 200 μL reaction was prepared, containing 8 μM CUL4/ROC1, 1.8 μM UBC12, 30 μM NEDD8 in 50 mM Tris-HCl pH 7.8, 150 mM NaCl, 2.5 mM MgCl$_2$, 2.5 mM ATP. 2x 30 μL samples were taken for SDS-PAGE, one was immediately mixed with 10 μL 4x SDS sample buffer, the other one incubated for 60 min at 25˚C. The reaction was initiated by addition of 0.7 μM APPBP1/UBA3, incubated at 25˚C, and 30 μL SDS-PAGE samples were taken after 1 min, 5 min, 10 min, 30 min and 60 min, immediately mixed with 10 μL 4x SDS sample buffer and boiled at 95˚C for 5 min. Samples were analysed by SDS-PAGE. Based on this test, the reaction was scaled up to 1 mL and incubated for 5 min at 25˚C. Reaction was quenched by addition of 5 mM TCEP and immediately loaded onto a Superdex 200 16/600 GF column (GE), equilibrated in 10 mM Tris-HCl pH 7.8, 150 mM NaCl, 4 mM MgCl$_2$, 0.5 mM TCEP at a flow rate of 1 mL/min. Peak fractions were analysed by SDS-PAGE, appropriate fractions were pooled and concentrated to ~20 mg/mL, flash-frozen in liquid nitrogen in small aliquots and stored at -80˚C.

### X-ray crystallography sample preparation, crystallisation, data collection and structure solution

**DDB1/DCAF1-CtD complex.** DDB1/DCAF1-CtD crystals were grown by the hanging drop vapour diffusion method, by mixing equal volumes (1 μL) of DDB1/DCAF1-CtD solution at 10 mg/mL with reservoir solution containing 100 mM Tri-Na citrate pH 5.5, 18% PEG 1000 and suspending over a 500 μl reservoir. Crystals grew over night at 18˚C. Crystals were cryo-protected in reservoir solution supplemented with 20% glycerol and cryo-cooled in liquid nitrogen. A data set from a single crystal was collected at Diamond Light Source (Didcot, UK) at a wavelength of 0.92819 Å. Data were processed using XDS [95] (S1 Table), and the structure was solved using molecular replacement with the program MOLREP [96] and available structures of DDB1 (PDB 3e0c) and DCAF1-CtD (PDB 4cc9) [50] as search models. Iterative

cycles of model adjustment with the program Coot [97], followed by refinement using the program PHENIX [98] yielded final $R/R_{free}$ factors of 22.0%/27.9% (S1 Table). In the model, 94.5% of residues have backbone dihedral angles in the favoured region of the Ramachandran plot, the remainder fall in the allowed regions, and none are outliers. Details of data collection and refinement statistics are presented in S1 Table. Coordinates and structure factors have been deposited in the PDB, accession number 6zue.

**DDB1/DCAF1-CtD/T4L-Vpr$_{mus}$ (1–92) complex.** The DDB1/DCAF1-CtD/Vpr$_{mus}$ complex was assembled by incubation of purified DDB1/DCAF1-CtD and HisSU-MO-T4L-Vpr$_{mus}$ (residues 1–92), at a 1:1 molar ratio, in a buffer containing 50 mM Bis-tris propane pH 8.5, 0.5 M NaCl, 4 mM $MgCl_2$, 0.5 mM TCEP, containing 1 mg of HRV-3C protease for HisSUMO-tag removal. After incubation on ice for 12 h, the sample was loaded onto a Superdex 200 16/600 GF column (GE), with a 1 mL GSH-Sepharose FF column (GE) connected in line. The column was equilibrated with 10 mM Bis-tris propane pH 8.5, 150 mM NaCl, 4 mM MgCl2, and 0.5 mM TCEP. The column flow rate was 1 mL/min. GF fractions were analysed by SDS-PAGE, appropriate fractions were pooled and concentrated to 4.5 mg/mL.

Crystals were prepared by the sitting drop vapour diffusion method, by mixing equal volumes (200 nL) of the protein complex at 4.5 mg/mL and reservoir solution containing 8–10% PEG 4000 (w/v), 200 mM $MgCl_2$, 100 mM HEPES-NaOH, pH 7.0–8.2. The reservoir volume was 75 μL. Crystals grew after at least 4 weeks of incubation at 4˚C. Crystals were cryo-protected in reservoir solution supplemented with 20% glycerol and cryo-cooled in liquid nitrogen. Data sets from two single crystals were collected, initially at BESSY II (Helmholtz-Zentrum Berlin, HZB) at a wavelength of 0.91841 Å, and later at ESRF (Grenoble) at a wavelength of 1 Å. Data sets were processed separately using XDS [95] and XDSAPP [99]. The structure was solved by molecular replacement, using the initial BESSY data set, with the program PHASER [100], and the following structures as search models: DDB1/DCAF1-CtD (this work) and T4L variant E11H (PDB 1qt6) [101]. After optimisation of the initial model and refinement against the higher-resolution ESRF data set, Vpr$_{mus}$ was placed manually into the density, using an NMR model of Vpr$_{HIV-1}$ (PDB 1m8l) [102] as guidance. Iterative cycles of model adjustment with the program Coot [97], followed by refinement using the program PHENIX [98] yielded final $R/R_{free}$ factors of 21.61%/26.05%. In the model, 95.1% of residues have backbone dihedral angles in the favoured region of the Ramachandran plot, the remainder fall in the allowed regions, and none are outliers. Details of data collection and refinement statistics are presented in S1 Table. Coordinates and structure factors have been deposited in the PDB, accession number 6zx9.

## Cryo-EM sample preparation and data collection

**Complex assembly.** Purified CUL4-NEDD8/ROC1, DDB1/DCAF1-CtD, GST-Vpr$_{mus}$ and rhesus macaque SAMHD1, 1 μM each, were incubated in a final volume of 1 mL of 10 mM Tris-HCl pH 7.8, 150 mM NaCl, 4 mM $MgCl_2$, 0.5 mM TCEP, supplemented with 1 mg of GST-3C protease. No additional GTP or dNTPs were added to ensure SAMHD1 was maintained in the apo- monomer-dimer form. After incubation on ice for 12 h, the sample was loaded onto a Superdex 200 16/600 GF column (GE), equilibrated with the same buffer at 1 mL/min, with a 1 mL GSH-Sepharose FF column (GE) connected in line. GF fractions were analysed by SDS-PAGE, appropriate fractions were pooled and concentrated to 2.8 mg/mL.

**Grid preparation.** 3.5 μl protein solution containing 0.05 μM CUL4-NEDD8/ROC1/DDB1/DCAF1-CtD/Vpr$_{mus}$/SAMHD1 complex and 0.25 μM UBCH5C-ubiquitin conjugate (S4A and S4B Fig) were applied to a 300 mesh Quantifoil R2/4 Cu/Rh holey carbon grid

(Quantifoil Micro Tools GmbH) coated with an additional thin carbon film as sample support and stained with 2% uranyl acetate for initial characterisation. For cryo-EM, a fresh 400 mesh Quantifoil R1.2/1.3 Cu holey carbon grid (Quantifoil Micro Tools GmbH) was glow-discharged for 30 s using a Harrick plasma cleaner with technical air at 0.3 mbar and 7 W. 3.5 μl protein solution containing 0.4 μM CUL4-NEDD8/ROC1/DDB1/DCAF1-CtD/Vpr$_{mus}$/ SAMHD1 complex and 2 μM UBCH5C-ubiquitin conjugate were applied to the grid, incubated for 45 s, blotted with a Vitrobot Mark II device (FEI, Thermo Fisher Scientific) for 1–2 s at 8˚C and 80% humidity, and plunged in liquid ethane. Grids were stored in liquid nitrogen until imaging.

**Cryo-EM data collection.** Initial negative stain and cryo-EM datasets were collected automatically for sample quality control and low-resolution reconstructions on a 120 kV Tecnai Spirit cryo-EM (FEI, Thermo Fisher Scientific) equipped with a F416 CMOS camera (TVIPS) using Leginon [103,104]. Particle images were then analysed by 2D classification and initial model reconstruction using SPHIRE [105], cisTEM [106] and Relion 3.07 [107]. These data revealed the presence of the complexes containing both DDB1/DCAF1-CtD/Vpr$_{mus}$ (core) and CUL4/ROC1 (stalk). High-resolution data was collected on a 300 kV Tecnai Polara cryo-EM (FEI, Thermo Fisher Scientific) equipped with a K2summit direct electron detector (Gatan) at a nominal magnification of 31000x, with a pixel size of 0.625 Å/px on the object scale. In total, 3644 movie stacks were collected in super-resolution mode using Leginon [103,104] with the following parameters: defocus range of 0.5–3.0 μm, 40 frames per movie, 10 s exposure time, electron dose of 1.25 e/Å$^2$/s and a cumulative dose of 50 e/Å$^2$ per movie.

## Cryo-EM computational analysis

Movies were aligned and dose-weighted using MotionCor2 [108] and initial estimation of the contrast transfer function (CTF) was performed with the CTFFind4 package [109]. Resulting micrographs were manually inspected to exclude images with substantial contaminants (typically large protein aggregates or ice contaminations) or grid artefacts. Power spectra were manually inspected to exclude images with astigmatic, weak, or poorly defined spectra. After these quality control steps the dataset included 2322 micrographs (63% of total). At this stage, the data set was picked twice and processed separately, to yield reconstructions of states-1, -2 and -3 (analysis 1) and of the core (analysis 2).

For **analysis 1**, particle positions were determined using cisTEMs Gaussian picking routine, yielding 959,155 particle images in total. After two rounds of 2D-classification, 227,529 particle images were selected for further processing (S4C and S4D Fig). Using this data, an initial model was created using Relion 3.07. The resulting map yielded strong signal for the core but only fragmented stalk density, indicating a large heterogeneity in the stalk-region within the data set. This large degree of compositional (+/- stalk) and conformational heterogeneity (movement of the stalk relative to the core) made the classification challenging. Accordingly, alignment and classification were carried out simultaneously. The first objective was to separate the data set into three categories: "junk", "core" and "core+stalk". Therefore, the stalk was deleted from the initial model using the "Eraser"-tool in Chimera [110]. This core-map was used as an initial model for the Tier 1 3D-classification with Relion 3.07 at a decimated pixel size of 2.5 Å/px. The following parameters were used: number of classes K = 6, T = 10, global step search = 7.5˚, number of iterations = 25. The classification yielded two classes containing the stalk (classes 3 and 5 containing 23% and 22% of the particle images, respectively) (S4D Fig). These particles were pooled and directed into Tier 2 3D-classification using the following parameters: number of classes K = 6, T = 10, global step search = 7.5˚, number of iterations = 25. Three of these classes yielded medium-resolution maps with interpretable features

(states-1, -2 and -3, S4D Fig). These three classes were refined individually using 3D autorefinement in Relion 3.07, resulting in maps with resolution ranging from 7.8 Å– 8.9 Å (S4E and S4F and S4G Fig).

For **analysis 2**, particle positions were determined using template matching with a filtered map comprising core and stalk using the software Gautomatch (https://www2.mrc-lmb.cam.ac.uk/research/locally-developed-software/zhang-software/). 712,485 particle images were found, extracted with Relion 3.07 and subsequently 2D-classified using cryoSPARC [111], resulting in 505,342 particle images after selection (S5A and S5B Fig). These particle images were separated into two equally sized subsets and Tier 1 3D-classification was performed using Relion 3.07 on both of them to reduce computational burden (S5B Fig). The following parameters were used: initial model = "core", number of classes K = 4, T = 10, global step search = 7.5˚, number of iterations = 25, pixel size 3.75 Å/px. From these, the ones possessing both core and stalk were selected. Classes depicting a similar stalk orientation relative to the core were pooled and directed into Tier 2 as three different subpopulations containing 143,172, 193,059 and 167,666 particle images, respectively (S5B Fig).

For Tier 2, each subpopulation was classified separately into 4 classes each. From these 12 classes, all particle images exhibiting well-defined densities for core and stalk were pooled and labelled "core+stalk", resulting in 310,801 particle images in total. 193,096 particle images representing classes containing only the core were pooled and labelled "core" (S5B Fig)

For Tier 3, the "core" particle subset was separated into 4 classes which yielded uninterpretable reconstructions lacking medium- or high-resolution features. The "core+stalk" subset was separated into 6 classes, with 5 classes containing both stalk and core (S5B Fig) and one class consisting only of the core with $Vpr_{mus}$ bound. The 5 classes with stalk showed similar stalk orientations as the ones obtained from analysis 1 (see above, S4 Fig), but refined individually to lower resolution as in analysis 1 and were discarded. However, individual refinement of the core-only tier 3 class yielded a 7.3 Å reconstruction (S5C and S5D Fig).

## Molecular visualisation, rigid body fitting, 3D structural alignments, rotation and interface analysis

Density maps and atomic models were visualised using Coot [97], PyMOL (Schrödinger) and UCSF Chimera [110]. Rigid body fits and structural alignments were performed using the program UCSF Chimera [110]. Rotation angles between extreme DDB1 BPB domain positions were measured using the DynDom server [112] (http://dyndom.cmp.uea.ac.uk/dyndom/runDyndom.jsp). Molecular interfaces were analysed using the EBI PDBePISA server [113] (https://www.ebi.ac.uk/msd-srv/prot_int/cgi-bin/piserver).

## Multiple sequence alignment

A multiple sequence alignment was calculated using the EBI ClustalOmega server [114] (https://www.ebi.ac.uk/Tools/msa/clustalo/), and adjusted manually using the program Gene-Doc [115].

## Cross-linking mass spectrometry (CLMS)

**Complex assembly.** Purified CUL4/ROC1, DDB1/DCAF1-CtD, GST-$Vpr_{mus}$ and rhesus macaque SAMHD1, 1 μM each, were incubated in a volume of 3 mL buffer containing 10 mM HEPES pH 7.8, 150 mM NaCl, 4 mM $MgCl_2$, 0.5 mM TCEP, supplemented with 1 mg GST-3C protease. After incubation on ice for 12 h, the sample was loaded onto a Superdex 200 16/600 GF column (GE), equilibrated with the same buffer, at a flow rate of 1 mL/min with a 1

mL GSH-Sepharose FF column (GE) connected in line. GF fractions were analysed by SDS-PAGE, appropriate fractions were pooled and concentrated to 6 mg/mL.

**Photo-crosslinking.**    The cross-linker sulfo-SDA (sulfosuccinimidyl 4,4′-azipentanoate) (Thermo Scientific) was dissolved in cross-linking buffer (10 mM HEPES pH 7.8, 150 mM NaCl, 4 mM $MgCl_2$, 0.5 mM TCEP) to 100 mM before use. The labelling step was performed by incubating 18 μg aliquots of the complex at 1 mg/mL with 2, 1, 0.5, 0.25, 0.125 mM sulfo-SDA, added, respectively, for an hour. The samples were then irradiated with UV light at 365 nm, to form cross-links, for 20 min and quenched with 50 mM $NH_4HCO_3$ for 20 min. All steps were performed on ice. Reaction products were separated on a Novex Bis-Tris 4–12% SDS−PAGE gel (Life Technologies). The gel band corresponding to the cross-linked complex was excised and digested with trypsin (Thermo Scientific Pierce) [116] and the resulting tryptic peptides were extracted and desalted using C18 StageTips [117]. Eluted peptides were fractionated on a Superdex Peptide 3.2/300 increase column (GE Healthcare) at a flow rate of 10 μL/min using 30% (v/v) acetonitrile and 0.1% (v/v) trifluoroacetic acid as mobile phase. 50 μL fractions were collected and vacuum-dried.

**CLMS acquisition.**    Samples for analysis were resuspended in 0.1% (v/v) formic acid, 3.2% (v/v) acetonitrile. LC-MS/MS analysis was performed on an Orbitrap Fusion Lumos Tribrid mass spectrometer (Thermo Fisher) coupled on-line with an Ultimate 3000 RSLCnano HPLC system (Dionex, Thermo Fisher). Samples were separated on a 50 cm EASY-Spray column (Thermo Fisher). Mobile phase A consisted of 0.1% (v/v) formic acid and mobile phase B of 80% (v/v) acetonitrile with 0.1% (v/v) formic acid. Flow rates were 0.3 μL/min using gradients optimized for each chromatographic fraction from offline fractionation, ranging from 2% mobile phase B to 55% mobile phase B over 90 min. MS data were acquired in data-dependent mode using the top-speed setting with a 3 s cycle time. For every cycle, the full scan mass spectrum was recorded using the Orbitrap at a resolution of 120,000 in the range of 400 to 1,500 m/z. Ions with a precursor charge state between 3+ and 7+ were isolated and fragmented. Analyte fragmentation was achieved by Higher-Energy Collisional Dissociation (HCD) [118] and fragmentation spectra were then recorded in the Orbitrap with a resolution of 50,000. Dynamic exclusion was enabled with single repeat count and 60 s exclusion duration.

**CLMS processing.**    A recalibration of the precursor m/z was conducted based on high-confidence (<1% false discovery rate (FDR)) linear peptide identifications. The re-calibrated peak lists were searched against the sequences and the reversed sequences (as decoys) of cross-linked peptides using the Xi software suite (v.1.7.5.1) for identification [119]. Final crosslink lists were compiled using the identified candidates filtered to <1% FDR on link level with xiFDR v.2.0 [120] imposing a minimum of 20% sequence coverage and 4 observed fragments per peptide.

**CLMS analysis.**    In order to sample the accessible interaction volume of the SAMHD1-CtD consistent with CLMS data, a model for SAMHD1 was generated using I-TASSER [121]. The SAMHD1-CtD, which adopted a random coil configuration, was extracted from the model. In order to map all crosslinks, missing loops in the complex structure were generated using MODELLER [122]. An interaction volume search was then submitted to the DisVis webserver [61] with an allowed distance between 1.5 Å and 22 Å for each restraint using the "complete scanning" option. The rotational sampling interval was set to 9.72˚ and the grid voxel spacing to 1Å. The accessible interaction volume was visualised using UCSF Chimera [110].

## Circular dichroism (CD) spectroscopy

In order to evaluate the secondary structure content of GST-tagged Vpr<sub>mus</sub>, and the double mutants R15E/R75E and W29A/A66W, CD spectroscopy was performed. Protein samples

were dialysed overnight at 4˚C against CD buffer (100 mM NaF, 10 mM $K_2HPO_4/KH_2PO_4$ mixture, pH 8.5), and protein concentration then adjusted to 0.2 mg/ml. For each sample, three replicate CD spectra were recorded in 1.0 nm steps over the range of 190–260 nm. Measurements were performed in 0.05 cm path-length quartz cuvettes (Hellma, Mühlheim, Germany) at 20˚C on a Chirascan spectropolarimeter (Applied Photophysics, London, UK). A reference spectrum of CD buffer was subtracted from the averaged protein sample spectrum. Measured ellipticity in millidegrees (m˚) was converted to molar ellipticity [Θ] according to equation $\Theta = m^{˚*}M/(10xLxC)$, where M is the average protein molecular weight, L is path length of the measurement cell, and C is the molar protein concentration. Molar ellipticity was plotted against wavelength using SigmaPlot version 14.0 (Systat Software Inc., San Jose, California, USA).

## Supporting information

**S1 Fig. Additional biochemical analysis of Vpr_mus-induced CRL4^DCAF1-CtD specificity redirection towards SAMHD1.** (**A**) Schematic view of the protein constructs used in biochemical analyses. BP–β-propeller domain, HD–histidine-aspartate domain, HTH–helix-turn-helix motif, SAM–sterile alpha motif. (**B**) SDS-PAGE analysis of GST-Vpr_mus. After treatment with 3C protease to remove the GST-tag (+3C) and GSH-Sepharose pull down to remove protease and tag, no Vpr_mus is present in the eluted fraction (S/N) indicating that it interacts non-specifically with the GSH-Sepharose beads and/or becomes insoluble after tag removal. (**C-E**) Analytical GF analysis of DDB1/DCAF1-CtD incubated with SAMHD1 (**C**), DDB1/DCAF1-CtD incubated with Vpr_mus (**D**) and SAMHD1 incubated with Vpr_mus (**E**). SDS-PAGE of the corresponding GF fractions is shown below each chromatogram.
(TIF)

**S2 Fig. Components, controls and uncropped SDS-PAGE images of *in vitro* ubiquitylation reactions.** (**A**) SDS-PAGE of individually purified protein components used in the *in vitro* ubiquitylation reactions. (**B, C**) Control reactions in the absence of indicated components. (**D-F**) Uncropped gels of reactions shown in Fig 1D and 1E and 1F and 1G. All reactions were incubated at 37˚C for the indicated times, stopped by addition of SDS sample buffer and separated on SDS-PAGE.
(TIF)

**S3 Fig. Detailed crystal structure analysis of the DDB1/DCAF1-CtD/Vpr_mus complex.** (**A**) Superposition of the Vpr_mus (green cartoon)/DCAF1-CtD complex with Vpx_sm (orange cartoon, PDB: 4cc9) [50], Vpx_mnd (blue cartoon, PDB: 5aja) [51] and Vpr_HIV-1 (light brown cartoon, PDB: 5jk7) [54]. Structures have been aligned with respect to their DCAF1 BP domains but only the DCAF1-CtD from the Vpr_mus complex is shown for clarity (grey cartoon and semi-transparent surface). (**B-G**) Details of the DCAF1-CtD/Vpr_mus interaction. (**B**) The structure of the complex is shown in the same orientation as Fig 2A, left panel. The insets (**C-G**) show individual interaction areas in more detail, Vpr_mus (green), Vpr_mus-bound DCAF1-CtD (grey) and apo-DCAF1-CtD (light blue). Selected amino acid residues, that make intermolecular interactions, are shown as sticks, and hydrogen bonds/electrostatic interactions as dashed red lines. Vpr_mus residues with asterisks are type-conserved within all Vpr/Vpx proteins. (**H**) Circular dichroism (CD) spectra of GST-Vpr_mus and GST-Vpr_mus variants R15E/R75E and W29A/A66W.
(TIF)

**S4 Fig. Cryo-EM analysis 1 of the CRL4-NEDD8^DCAF1-CtD/Vpr_mus/SAMHD1 complex.** (**A**) *In vitro* neddylation of CUL4/ROC1. Protein was mixed with purified neddylation-E1

(APPBP1/UBA3 heterodimer), E2 (UBC12) and NEDD8. The reaction was incubated at 25˚C, samples were taken at indicated times, stopped by addition of SDS sample buffer and separated on SDS-PAGE. (**B**) GF analysis of the CUL4-NEDD8/ROC1/DDB1/DCAF1-CtD/Vpr$_{mus}$/ SAMHD1 complex with pooled fractions indicated. A 5x molar excess of UBCH5C-ubiquitin was added before plunge-freezing for cryo-EM experiments, in an attempt to stabilise the assembly. However, no density in any of the reconstructions could be assigned to UBCH5C-ubiquitin, indicating low binding affinity and/or heterogeneity in its mode of binding. (**C**) 2D class averages depicting either "core" or "core+stalk" classes of analysis 2. (**D**) 3D sorting tree after 2D classification. Conformational states-1, -2 and -3 are indicated. (**E**) Local resolution and Euler distribution of states-1, -2 and -3. (**F**) FSC curves for state-1, -2 and -3 reconstructions. (**G**) Side-by-side comparison of state-1, -2 and -3 reconstructions, coloured as in Fig 3. Molecular models of the DDB1/DCAF1-CtD/Vpr$_{mus}$ crystal structure and CUL4/ROC1 (PDB 2hye) [15] have been fitted as rigid bodies into the volumes and are shown as cartoons. DDB1/ DCAF1-CtD/Vpr$_{mus}$ is coloured as in Fig 4, CUL4 is coloured yellow and ROC1 cyan. All states show additional density corresponding to SAMHD1-CtD, indicated by the red arrows. (**H**) Superposition of the neddylated CUL5 C-terminal WHB domain (black cartoon, PDB 3dqv) [56] on the CUL4 WHB (PDB 2hye), coloured as in **A**. Respective lysine residues, which are covalently modified with NEDD8, are indicated. (**I, J**) Detailed view of state-1 (I) and state-3 (J) cryo-EM density. Red arrows indicate contacts between CUL4A (orange cartoon) and DDB1 BPA/BPC/CtD (blue cartoon).
(TIF)

**S5 Fig. Cryo-EM analysis 2 and CLMS analysis of the CRL4(-NEDD8)$^{DCAF1-CtD}$/ Vpr$_{mus}$ /SAMHD1 complex.** (**A**) 2D class averages depicting either CRL4-NEDD8$^{DCAF1-CtD}$/Vpr$_{mus}$/ SAMHD1 "core" or "core+stalk" classes of analysis 1. (**B**) Sorting tree after 2D classification. In Tier 3, the core reconstruction was identified, containing 106,747 particle images (red box). (**C**) Local resolution of the core reconstruction after refinement, indicating a resolution range from 6.5 Å in the hydrophobic interior of DDB1 to 10.5Å in the DDB1 BPB domain. Below, the Euler distribution is shown. (**D**) FSC curve of the core reconstruction after refinement. (**E**) Upper panel: CRL4$^{DCAF1-CtD}$/Vpr$_{mus}$/SAMHD1 cross-links, identified by CLMS, mapped on molecular models representing state-1, -2 and -3. Satisfied crosslinks (<25 Å) are coloured yellow, violated crosslinks red. Red arrows indicate a subset of cross-links between DDB1 and CUL4, whose distance restraints are satisfied in state-1, and increasingly violated in states-2 and -3. Lower panel: circle plot of CLMS data for states-1, -2 and -3, using the same colour scheme as in the upper panel. Grey lines represent crosslinks between residues that are not present in the molecular models. Only crosslinks between subunits are displayed.
(TIF)

**S6 Fig. Multiple sequence alignment of Vpr/Vpx proteins, detailed structural comparison between Vpr$_{mus}$ and Vpr$_{HIV-1}$.** (**A**) Sequence alignment of indicated Vpr and Vpx proteins. Helices are indicated by the boxes above the amino acid sequences for Vpr$_{HIV-1}$ (pink) and Vpr$_{mus}$ (green). Vpr$_{HIV-1}$ side chains involved in UNG2-binding are indicated with pink asterisks. Vpr$_{mus}$ side chains putatively involved in SAMHD1-CtD-binding are indicated by green asterisks. Vpx$_{sm}$ side chains targeting SAMHD1-CtD are indicated with orange asterisks, and Vpx$_{mnd2}$ side chains contacting N-terminal SAMHD1 domains are highlighted with blue asterisks. Red symbols mark Vpr$_{mus}$ side chains involved in DCAF1-CtD-binding. The non-outlined symbols indicate DCAF1-CtD-contacting side chains unique to Vpr$_{mus}$, and dashes show DCAF1-binding side chains, which are in contact with DCAF1-CtD in other Vpr/Vpx structures, but not in Vpr$_{mus}$. Grey asterisks mark Vpr/Vpx side chains involved in zinc coordination. (**B**) Structural alignment of Vpr$_{HIV-1}$ (PDB 5jk7 [54], light brown) in complex with

UNG2 and Vpr$_{mus}$ (green). Protein backbone is shown in cartoon representation. For clarity, only the DNA-intercalating loop of UNG2 is shown (pink), which inserts into a hydrophobic pocket created by the Vpr$_{HIV-1}$ helix bundle. Note the steric clash between UNG2 side chain L272 and Vpr$_{mus}$ residue W48 in the structural superposition. (**C**) Alternative view of the structural alignment of Vpr$_{HIV-1}$ (light brown) in complex with UNG2 (pink) and Vpr$_{mus}$ (green). Note the steric clash between UNG2 and the extended Helix-3 of Vpr$_{mus}$. (TIF)

**S1 Table. X-ray data collection and refinement statistics.** *Numbers in parentheses account for the high-resolution shell, **defined in [124].
(PDF)

**S2 Table. Oligonucleotide primer sequences.**
(PDF)

**S3 Table. Expression constructs.**
(PDF)

## Acknowledgments

We thank the MPI-MG for granting access to the TEM instruments of the microscopy and cryo-EM service group. We thank Dr. Manfred Weiss and the scientific staff of the BESSY-MX (Macromolecular X-ray Crystallography)/Helmholtz Zentrum Berlin für Materialien und Energie at beamlines BL14.1, BL14.2, and BL14.3 operated by the Joint Berlin MX-Laboratory at the BESSY II electron storage ring (Berlin-Adlershof, Germany) as well as the scientific staff of the ESRF (Grenoble, France) at beamlines ID30A-3, ID30B, ID23-1, ID23-2, and ID29 for continuous support. We acknowledge Diamond Light Source (Didcot, UK) for access and support of the synchrotron beamline I04 and cryo-EM facilities at the UK's national Electron Bio-imaging Centre (eBIC). Furthermore, the authors acknowledge the North-German Supercomputing Alliance (HLRN) and the HPC for Research cluster of the Berlin Institute of Health for providing HPC resources. We are grateful to Prof. Udo Heinemann and Jianhui Wang (Max-Delbrück-Centrum, Berlin, Germany) for access to and assistance during CD spectroscopy. The pHisSUMO plasmid was a generous gift from Dr. Evangelos Christodoulou (The Francis Crick Institute, London, UK). The rhesus macaque SAMHD1 cDNA template was a generous gift from Prof. Michael Emerman (Fred Hutchinson Cancer Research Center, Seattle, USA). Recombinant BAC10:1629KO bacmid was a generous gift from Prof. Ian Jones (University of Reading, UK). pAcGHLT-B-DDB1 was a gift from Prof. Ning Zheng (Addgene plasmid 48638). pET28-mE1 was a gift from Prof. Jorge Eduardo Azevedo (Addgene plasmid 32534).

## Author Contributions

**Conceptualization:** Sofia Banchenko, Ferdinand Krupp, Jörg Bürger, Andrea Graziadei, Francis J. O'Reilly, Ludwig Sinn, Juri Rappsilber, Christian M. T. Spahn, Thorsten Mielke, Ian A. Taylor, David Schwefel.

**Data curation:** Sofia Banchenko, Ferdinand Krupp, Jörg Bürger, Andrea Graziadei, Francis J. O'Reilly, Ludwig Sinn, David Schwefel.

**Formal analysis:** Sofia Banchenko, Ferdinand Krupp, Christine Gotthold, Jörg Bürger, Andrea Graziadei, Francis J. O'Reilly, Ludwig Sinn, David Schwefel.

**Funding acquisition:** Juri Rappsilber, Christian M. T. Spahn, Thorsten Mielke, Ian A. Taylor, David Schwefel.

**Investigation:** Sofia Banchenko, Ferdinand Krupp, Christine Gotthold, Jörg Bürger, Andrea Graziadei, Francis J. O'Reilly, Ludwig Sinn, Olga Ruda, David Schwefel.

**Methodology:** Sofia Banchenko, Ferdinand Krupp, Christine Gotthold, Jörg Bürger, Andrea Graziadei, Francis J. O'Reilly, Ludwig Sinn, Olga Ruda, Juri Rappsilber, Christian M. T. Spahn, Thorsten Mielke, Ian A. Taylor, David Schwefel.

**Project administration:** Juri Rappsilber, Christian M. T. Spahn, Thorsten Mielke, Ian A. Taylor, David Schwefel.

**Resources:** Sofia Banchenko, Christine Gotthold, Jörg Bürger, Andrea Graziadei, Francis J. O'Reilly, Ludwig Sinn, Juri Rappsilber, Christian M. T. Spahn, Thorsten Mielke, Ian A. Taylor, David Schwefel.

**Software:** Sofia Banchenko, Andrea Graziadei, Francis J. O'Reilly, Ludwig Sinn, Juri Rappsilber, Thorsten Mielke.

**Supervision:** Sofia Banchenko, Christine Gotthold, Juri Rappsilber, Christian M. T. Spahn, Thorsten Mielke, Ian A. Taylor, David Schwefel.

**Validation:** Sofia Banchenko, Ferdinand Krupp, David Schwefel.

**Visualization:** Sofia Banchenko, Ferdinand Krupp, Andrea Graziadei, Francis J. O'Reilly, David Schwefel.

**Writing – original draft:** David Schwefel.

**Writing – review & editing:** Sofia Banchenko, Ferdinand Krupp, Andrea Graziadei, Juri Rappsilber, Christian M. T. Spahn, Thorsten Mielke, Ian A. Taylor, David Schwefel.

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
