## [Decision Letter · Decision Letter 0]

14 May 2021

Dear Dr. Schwefel,

Thank you very much for submitting your manuscript "Structural insights into Cullin4-RING ubiquitin ligase remodelling by Vpr from simian immunodeficiency viruses" for consideration at PLOS Pathogens. As with all papers reviewed by the journal, your manuscript was reviewed by members of the editorial board and by several independent reviewers. The reviewers appreciated the attention to an important topic. Based on the reviews, we are likely to accept this manuscript for publication, providing that you modify the manuscript according to the review recommendations.

Authors are encouraged to consider the suggestion from reviewer #1 to include the sequence alignment (currently in Figure S6) in Figure 6, and should address comments raised by reviewer #2 in the discussion section of the manuscript

Sincerely,

Gilda Tachedjian, Ph.D.

Associate Editor

PLOS Pathogens

Richard Koup

Section Editor

PLOS Pathogens

Kasturi Haldar

Editor-in-Chief

PLOS Pathogens

orcid.org/0000-0001-5065-158X

Michael Malim

Editor-in-Chief

PLOS Pathogens

orcid.org/0000-0002-7699-2064

Authors are encouraged to consider the suggestion from reviewer #1 to include the sequence alignment (currently in Figure S6) in Figure 6, and should address comments raised by reviewer #2 in the discussion section of the manuscript

Reviewer Comments (if any, and for reference):

Reviewer's Responses to Questions

**Part I - Summary**

Reviewer #1: The Vpr/Vpx accessory proteins of HIV/SIV hijack the same CRL4-DCAF1 E3 ubiquitin ligase to mediate degradation of multiple cellular factors to facilitate infection. Previous structural studies with SIVsm Vpx, SIVmnd Vpx, and HIV-1 Vpr by this group and others highlighted that these viral proteins essentially fold and interact with DCAF1 in the same manner, but recruit structurally diverse proteins. Banchenko et al., now detail the molecular architecture of SIVmus Vpr interaction with the CRL4-DCAF1 E3 by conducting integrative structural analyses with X-ray crystallography, cryo-electron microscopy, and cross-linking mass spectrometry. The data show that SIVmus Vpr utilizes the common modality to interact with DCAF1, while it employs unique interface for recruitment of the SAMHD1 C-terminus. The structural data are well supported by biochemical and mutational analyses. The paper is well written with substantial amount of data, which support the major findings. This should be of interest to HIV community as well as ubiquitin field.

Reviewer #2: This manuscript reveals the structural insights about the molecular interactions among Cullin4-RING ubiquitination complex (DCAF1), host SAMHD1 protein and an unique SIV Vpr protein harboring SAMHD1 degradation capability, called hybrid Vpr. Both X-ray crystallography and CryoEM methods were employed to gain the understanding of the molecular interaction mechanisms engineered by the hybrid Vpr during SAMHD1 proteosomal degradation. Revealing that this hybrid Vpr establishes different interaction strategies to SAMHD1, compared to well characterized Vpx proteins that degrade host SAMHD1, confirms that strategies adapted by various lentiviruses for counteracting host SAMHD1 proteins were differentially established throughout evolution and escape from the SAMHD1-mediated anti-viral selective pressures. Both methodologies and data interpretations were properly conducted. However, a few issues can be addressed for clarifications and better presentations.

**Part II – Major Issues: Key Experiments Required for Acceptance**

Reviewer #1: None

Reviewer #2: No major issues.

**Part III – Minor Issues: Editorial and Data Presentation Modifications**

Reviewer #1: Sequence alignment of HIV-1 Vpr, SIVmnd Vpx, SIVsm Vpx, and SIVmus Vpr can be incorporated with Fig. 6, highlighting utilization of different residues at distinct location for specific substrate recruitment, and common or the same residue types for DCAF1 interaction based on the published and current data set. S6 Fig contains such information, however it is too crowded to appreciate the message.

Reviewer #2: 1) It is not clear whether dGTP was used during structural analyses. Plus, it is also not clear which form of SAMHD1 is expected in the models presented in Figure 7. If dGTP was not used in any of both biochemical and structural analyses, SAMHD1 dimers could be the form that existed during the analyses. This could be an important issue, particularly because the molecular behaviors of the C-T domain could be significantly different between tetramers (with dGTP) and dimers (without dGTP).

2) It is likely that there is a significant molar ratio difference in infected macrophages between SAMHD1 and Vpr (or Vpx) carried by an infecting virion: SAMHD1 is highly abundant in macrophages whereas Vpr molecule numbers carried by a single virion should be much limited per cell. Therefore, it is not clear whether the model presented in Figure 7 can postulate any scenarios that may explain the degradation dynamics imposed by the vastly different molar ratios between SAMHD1 and Vpr per infected macrophage. Discussion on this issue will be helpful.

3) It remains somewhat unclear whether free Vpr captures SAMHD1 first and then binds to DACF1 in the complex, or Vpr binds to DCAF1 in the complex first and then captures SAMHD1. Possibly the SAMHD1-DCAF1 binding could be more stationary while Vpr binding to SAMHD1 is rather transient. Discussion on this issue will be helpful.

4) The Vpr proteins that carry anti-SAMHD1 activity has been considered as "ancient" Vpr before the gene duplication to create Vpx, leading to its loss of anti-SAMHd1 function. This point needs to be addressed in Introduction for giving a better definition of "hybrid' Vpr.

PLOS authors have the option to publish the peer review history of their article (what does this mean?). If published, this will include your full peer review and any attached files.

Reviewer #1: No

Reviewer #2: No

Figure Files:

Data Requirements:

Reproducibility:

References:

---

## [Editor Report · Decision Letter 1]

2 Jul 2021

Dear Dr. Schwefel,

We are pleased to inform you that your manuscript 'Structural insights into Cullin4-RING ubiquitin ligase remodelling by Vpr from simian immunodeficiency viruses' has been provisionally accepted for publication in PLOS Pathogens.

Best regards,

Gilda Tachedjian, Ph.D.

Associate Editor

PLOS Pathogens

Richard Koup

Section Editor

PLOS Pathogens

Kasturi Haldar

Editor-in-Chief

PLOS Pathogens

orcid.org/0000-0001-5065-158X

Michael Malim

Editor-in-Chief

PLOS Pathogens

orcid.org/0000-0002-7699-2064
---

## [Editor Report · Acceptance letter]

28 Jul 2021

Dear Dr. Schwefel,

We are delighted to inform you that your manuscript, "Structural insights into Cullin4-RING ubiquitin ligase remodelling by Vpr from simian immunodeficiency viruses," has been formally accepted for publication in PLOS Pathogens.

Best regards,

Kasturi Haldar

Editor-in-Chief

PLOS Pathogens

orcid.org/0000-0001-5065-158X

Michael Malim

Editor-in-Chief

PLOS Pathogens

orcid.org/0000-0002-7699-2064